# Ticks Infesting Dogs in Khyber Pakhtunkhwa, Pakistan: Detailed Epidemiological and Molecular Report

**DOI:** 10.3390/pathogens12010098

**Published:** 2023-01-06

**Authors:** Jehan Zeb, Baolin Song, Haytham Senbill, Muhammad Umair Aziz, Sabir Hussain, Munsif Ali Khan, Ishtiaq Qadri, Alejandro Cabezas-Cruz, José de la Fuente, Olivier Andre Sparagano

**Affiliations:** 1Department of Infectious Diseases and Public Health, Jockey Club College of Veterinary Sciences, City University of Hong Kong, Hong Kong SAR 999077, China; 2Department of Applied Entomology and Zoology, Faculty of Agriculture, Alexandria University, Alexandria 21545, Egypt; 3Vector-Borne Diseases Control Unit, District Health Office, Abbottabad 22010, Pakistan; 4Department of Biology, Faculty of Science King Abdulaziz University, Jeddah 21589, Saudi Arabia; 5ANSES, INRAE, Ecole Nationale Vétérinaire d’Alfort, UMR BIPAR, Laboratoire de Santé Animale, F-94700 Maisons-Alfort, France; 6SaBio, Instituto de Investigaci’on en Recursos Cineg´eticos IREC (CSIC-UCLM-JCCM), Ronda de Toledo 12, 13005 Ciudad Real, Spain; 7Department of Veterinary Pathobiology, College of Veterinary Medicine, Oklahoma State University, Stillwater, OK 74078, USA

**Keywords:** Hyalomma excavatum, Rhipicephalus sanguineus s.l., Rhipicephalus turanicus s.s., cox1, 16S rRNA, ticks, dogs, Khyber Pakhtunkhwa, Pakistan

## Abstract

Ticks and tick-borne diseases are considered a major challenge for human and animal health in tropical, sub-tropical, and temperate regions of the world. However, only scarce information is available on the characterization of tick species infesting dogs in Pakistan. In this study, we present a comprehensive report on the epidemiological and phylogenetic aspects of ticks infesting dogs in Pakistan using the mitochondrial markers i.e. Cytochrome c oxidase subunit 1 (*cox1*) and 16S ribosomal RNA (*16S rRNA*) nucleotide sequences. A total of 300 dogs were examined and 1150 ixodid ticks were collected across central Khyber Pakhtunkhwa, Pakistan. The morpho-molecular characterization of hard ticks revealed the presence of two ixodid tick genera on dogs, i.e., *Hyalomma* and *Rhipicephalus*, including six tick species viz. *Hyalomma dromedarii* (15.9%)*, Hyalomma excavatum* (3%), *Rhipicephalus sanguineus s.l.* (41.3%)*, Rhipicephalus turanicus s.s.* (28.7%), *Rhipicephalus haemaphysaloides* (10.2%), and *Rhipicephalus microplus* (2%). The total prevalence of tick infestation in dogs was 61%. The district with the highest tick prevalence rate in dogs was Mardan (14.7%), followed by Peshawar (13%), Swabi (12%), Charsadda (11%), and Malakand (10.3%), respectively. Risk factors analysis indicated that some demographic and host management-associated factors such as host age, breed, exposure to acaricides treatment, and previous tick infestation history were associated with a higher risk of tick infestation on dogs. This is the first molecular report confirming the infestation of *Hyalomma* and *Rhipicephalus* tick species in the dog population from the study area. The present study also reported a new tick–host association between *Hy. excavatum*, *Hy. dromedarii*, and dogs. Phylogenetic analysis revealed that *cox1* partial nucleotide sequences of *Hy. excavatum* in our dataset were 100% identical to similar tick specimens identified in Turkey, and those of *Hy. dromedarii* were identical to tick specimens from Iran. Whereas, *Rh. haemaphysaloides* and *Rh. microplus’ cox1* partial nucleotide sequences were identical to sequences previously published from Pakistan. *Rhipicephalus turanicus s.s.* ‘s *cox1* isolates from the present study were 99.8–100% identical to Pakistani-reported isolates, and those of *Rh. sanguineus s.l.* were 100% identical to Chinese specimens. Results on the genetic characterization of ticks were further confirmed by *16S rRNA* partial nucleotide sequences analysis, which revealed 100% identity between the tick isolates of this study and those of *Hy. excavatum* reported from Turkey; *Hy. dromedarii* specimens reported from Senegal; *Rh. haemaphysaloides*, *Rh. microplus*, and *Rh. turanicus s.s.*, previously published from Pakistan, and *Rh. sanguineus s.l.*, published from China. Furthermore, phylogenetic analysis showed that the *Rh. sanguineus s.l.* isolates of this study clustered with specimens of the tropical lineage with 7.7–10% nucleotide divergence from the specimens of the temperate lineage. Further molecular works need to be performed throughout Pakistan to present a more detailed map of tick distribution with information about dog host associations, biological characteristics, and pathogen competence.

## 1. Introduction

Ticks (Acari: Ixodidae) are important hematophagous ectoparasites infesting humans and animals including dogs in different agroecological zones of the world [1]. These ectoparasites can cause direct damage to the host by sucking large quantities of blood, resulting in anemia and indirectly increasing the possibility of secondary microorganismal infections that result in skin pathologies, i.e., abscesses, etc. [2,3]. Ticks can also cause paralysis, and more importantly, they harbor and transmit different pathogens affecting animals and humans alike [3]. Ticks are considered second to mosquitoes as vectors of human diseases but are the most important vectors of pathogens affecting animal health globally [4].

Dogs are cosmopolitan companion animals and are frequently found in human dwellings [5]. Thus, ticks carried by dogs can infest humans and transmit zoonotic diseases [6]. Dogs and other canines are preferred hosts of several tick genera and species, including among others, *Ixodes* (*I. affinis*, *I. arboricola*, *I. canisuga*, *I. kaiseri*, *I. kazakstani*), *Amblyomma* (*A. aureolatum*, *A. gervaisi*, *A. neumanni*, *A. ovale*, *A. parvum*), *Dermacentor* (*D. compactus*, *D. niveus*, *D. reticulatus*, *D. taiwanensis*), *Haemaphysalis* (*Hae. Anomala*, *Hae. Asiatica*, *Hae. Bispinosa*, *Hae. Camicasi*), *Hyalomma* (*Hy. albiparmatum*, *Hy. hussaini*, *Hy. impeltatum*, *Hy. rufipes*), and *Rhipicephalus* (*Rh. sanguineus s.l.*, *Rhipicephalus turanicus s.s., Rh. compositus*, *Rh. geigyi*, *Rh. haemaphysaloides*, *Rh. simus*) [7]. Of all the above tick species, *Rh. sanguineus s.l.*, the brown dog tick, is considered the most common tick infesting dogs in both urban and rural areas [8,9]. The cosmopolitan distribution and wide range of hosts enable *Rh. sanguineus s.l.* to complete its life cycle within human dwellings through pet dogs. Humans are incidental hosts of *Rh. sanguineus s.l.* and exposed to a wide variety of pathogens transmitted by this tick species [10]. Uninformative morphological description of the type specimen and loss of the original holotype of the *Rh. sanguineus* complex, described by Latreille in 1806, led to a massive conflict in its morpho-taxonomy [11,12]. Alternatively, the *Rh. sanguineus* complex has been divided into different lineages using molecular phylogenetics: *viz.* tropical, proposed to be referred to as *Rh. linnaei* by [9]; temperate, designated to be the actual *Rh. sanguineus s.s.* [12] and southeastern Europe lineage, in addition to closely similar species clades (*Rh. turanicus s.s.*) [13].

Several risk factors have been associated with a tick infestation in livestock/companion animals. For instance, host demography (e.g., age, gender, and breed) and management practice (e.g., acaricidal use, dogs roaming) associated attributes have been shown to influence tick distribution in different parts of the world [14]. Only a few studies in Pakistan have identified the risk factors linked to tick infestation on livestock farms and companion animals [15,16,17]. On the other hand, accurate identification of tick species, which mostly relies on morphological keys, is necessary when devising tick control strategies [11,18]. However, morphological identification of ticks requires expertise and might be challenging in the case of engorged or physically damaged specimens [19]. Therefore, alternative methods, such as the use of mitochondrial markers, *cox1,* and *16S rRNA*, have been approved their suitability for molecular identification and inferring phylogenies of the tick species, and hence, lead to the development of the DNA barcoding system for ticks [20,21,22,23].

Except for one morphological study that listed the three main genera of ixodid ticks (*Haemaphysalis*, *Hyalomma*, and *Rhipicephalus*) on dogs [17], no research work has yet been conducted in Pakistan to examine the molecular characteristics, genetic diversity, and epidemiology of hard ticks infesting dogs. To fill this crucial knowledge gap, the current study was designed to identify ticks morpho-molecularly in a randomized sampling of pet dogs to assess their prevalence, spatio-temporal distribution patterns, associated risk factors, and molecular phylogenies across the study area.

## 2. Materials and Methods

### 2.1. Ethical Consent

This experimental work was performed according to the Pakistan Veterinary Association (PVA) Manual of Animal Use. The ethical approval for this study was granted by the ethical committee on animal care and use (CVSAH/FCLS/AWKUM/2021/228) at the College of Veterinary Sciences and Animal Husbandry, Abdul Wali Khan University Mardan Pakistan. 

### 2.2. Study Location

Pakistan is predominantly an agricultural country and mainly divided into five agroecological zones according to remote sensing climate compound index-based climatic/aridity data analysis (hyper-arid, arid, humid, wetland, and cold drought) [24,25]. The study area (District Charsadda 34.1495° N, 71.7428° E; District Mardan 34.1937969° N, 72.0451467° E; District Malakand 34.5030° N, 71.9046° E; District Peshawar 33.9437° N, 71.6199° E; and District Swabi 34.0719° N, 72.4732° E) is located almost in the center of the Khyber Pakhtunkhwa Province of Pakistan (Figure 1) with dynamic environmental conditions, which in turn affect the distribution pattern of tick and tick-borne diseases [24,26]. Different types of vegetation can be found in this area, which may provide shelter to the growing ticks. In the study area, summers begin in the middle of April and peak in May and early June. The study area experiences seasonal monsoons (heavy rainfall in August), which cause a drop in daily mean temperatures and an increase in relative humidity, which in turn favor tick survival. On the other hand, winters are short, dry, and foggy, with considerable precipitation in January and February. The minimum and maximum average temperature recorded in the study area are 5.0 ± 2.5 °C and 40.2 ± 5.8 °C, while the minimum and maximum mean relative humidity recorded in the study region are 17.7 ± 2.5 and 65 ± 3.6, respectively [24,25,26].

### 2.3. Study Design, Tick Sampling Strategy, and Morpho-taxonomic Identification

The current study was carried out between January 1, 2021 and August 30, 2021 to investigate the diversity of ticks infesting dog population of the study area. Dogs of willing pet owners were included in this study. A simple random sampling strategy was adopted for dog examination and tick sample collection. A questionnaire was designed to collect information about the dogs included in the study. Dog owners were asked for the age, gender, breed, and roaming range of the dogs, as well as the occurrence and history of tick infestation and use of acaricides on these companion animals. 

A total of 300 dogs from different villages and towns (number of sampling spots = 40) of the study area were examined for tick infestation. Tick samples were collected following standard tick collection methods without any physical stress/harm to the dogs. For examination, dogs were restrained using a mouth gag/muzzle with the owner or handler’s aid so that the entire body could be thoroughly checked for ticks’ presence. Ticks were collected thoroughly from different dogs to ensure a reliable estimation of tick prevalence in the canine host population. Collected ticks were preserved in 70% ethanol and shipped via FedEx courier to the public health laboratory at the City University of Hong Kong for further investigations. Only unfed ticks were selected for morphological identification under a stereo zoom microscope (Olympus^®^, Tokyo, Japan) and identified up to the species level using standard identification keys [27,28]. The dogs were classified into three age-wise categories: puppy (<1 year), juvenile (1-3 years), and adult (>3 years). Two dog breeds were included in the study, i.e. long-haired and short-haired [29].

### 2.4. Tick Genomic DNA Extraction and Target Genes Amplification

A total of 120 morphologically identified hard ticks were selected (20 specimens of each species) for genomic DNA extraction. The DNA was extracted from each tick separately using the QIAamp^®^ DNA Mini Kit (Qiagen, Hilden, Germany) per the manufacturer’s instructions. A spectrophotometer (NanoDrop^TM^ Thermo Scientific^TM^, Waltham, Massachusetts, USA) was used to quantify DNA in each sample. All DNA samples were stored at deep freezing (– 80°C) for further downstream analysis. The *cox1* and *16S rRNA* genes of the ticks were amplified using tick-specific primer sets (Table 1) by Polymerase Chain Reactions, as previously described [23,26,30]. Briefly, the PCR reactions were carried out in a 30 µL volume of reaction mixture, including 2.5 µL of genomic DNA, 1 µL of each primer (forward and reverse, 10 pmol), 10.5 µL of PCR grade water, and 15 µL of master mix. PCR products were run on 2% agarose gel stained with ethidium bromide to visualize the *cox1* and *16S rRNA* amplicons under UV light in a gel documentation system (Bio-Rad Laboratories, Hercules, CA, USA).

### 2.5. Amplicons Purification and Sequencing

The *cox1* and *16S rRNA* amplicon samples were sent to BGI Tech Solutions (Hong Kong Co. limited, SAR China) for purification and sequencing. All the samples were sequenced and the query sequences in the dataset were edited for trimming and removal of unnecessary nucleotides at terminal ends. The query dataset was aligned in MEGA7 [33] and blasted against the NCBI GenBank database for complete alignment with the global isolates of the relevant tick species. For downstream bioinformatics analysis, all the relevant subject sequences with query coverage of 99.35–100% were downloaded and stored as separate datasets. The *cox1* and *16S rRNA* partial nucleotide query sequences were submitted to NCBI GenBank (*cox1*: ON911972- ON911982; *16S rRNA*: ON921112- ON921125).

### 2.6. Phylogenetic Analysis

To infer the phylogenetic relationship of the identified ticks, *cox1* and *16S rRNA* partial nucleotide sequences-based phylogenetic trees were constructed using Neighbor-joining (NJ) and Maximum likelihood (ML) algorithms with rapid bootstrapping of 1000 replicates in MEGA7 software [33]. The best-fit model of the sequences’ evolution was chosen based on the lowest Bayesian Information Criterion (BIC), Corrected Akaike Information Criterion (AICc), and Maximum Likelihood (*Inl*) values. The General Time Reversible model [34] was used for the *cox1*-based phylogenetic tree and the Tamura three-parameter model [35] for the *16S rRNA*-based phylogenetic tree. Neighbor-joining algorithm-based phylograms were rooted to determine the direction of evolution of the collected tick species, while the ML-algorithm-based trees were constructed without rooting for *Rh. sanguineus s.l.* isolates to determine their claustration and evolutionary relatedness, with tropical and temperate lineages of the brown dog tick, published globally.

### 2.7. Statistical Analysis of Empirical Data

The datasets from the present study were analyzed statistically using R software version 3.5.1 (R Development Core Team). The tick prevalence rate was analyzed by chi-square statistic to determine any significant association between host demographic or management factors and tick prevalence. Demographic and host management/environ-mental attributes were subjected to univariate and multivariate logistic regression analyses to predict their role as significant risk factors facilitating tick infestation in a canine population (dogs) across the study area. A confidence interval (CI) of 95% and *p* < 0.05 was considered statistically significant in all analyses.

## 3. Results

### 3.1. Host Demographic Profile

Among the 300 examined dogs, in total, 123 (41%) were puppies, 99 (33%) were juveniles, and 78 (26%) were adults. Gender-based analysis showed that 184 (61.4%) were female and 116 (38.6%) were males. In addition, the host population had 182 (60.6%) long-haired breed dogs while 118 (39.3%) were short-haired breed dogs. Among the examined dogs, 201 (67.0%) were free-roaming (allowed to roam freely outside the home/territory but have ownership) and 99 (33.0%) were non-roaming (Table 2).

### 3.2. Prevalence and Distribution of Ixodid Ticks

#### 3.2.1. Total and District-wise Prevalence of Ticks

The total prevalence of tick infestation was 61% in the dog population across the study area. The district-wise prevalence rate showed that it was high in dogs from Mardan (14.7%), followed by Peshawar (13%), Swabi (12%), Charsadda (11%), and Malakand (10.3%), respectively (Table 3).

#### 3.2.2. Tick Prevalence with Respect to Host Demography

Ixodid tick prevalence also varied with respect to the host’s demographic profile. Among the different age groups of the host, puppy dogs had significantly (*p* <0.05) higher tick infestation (34.7%) than juveniles (18.0%) and adults (8.3%). Gender-based prevalence showed that female dogs (42.0%) had higher tick infestation (*p* <0.05) as compared to male counterparts (18.7%). A comparison of the different host breeds showed that short-haired dogs were found to be more often infested with ticks (42.0%) as compared to long-haired breeds (19.0%), but this association was not statistically significant (*p* = 0.05) (Table 4).

#### 3.2.3. Tick Prevalence with Respect to Host Management Practices

The ixodid tick’s prevalence rate showed variation with respect to the host management practices by the owners. It was significantly higher (*p* <0.05) in dogs with no acaricidal application (43.3%) as compared to dogs with irregular (12.7%) and regular use of acaricides (5.0%). The host’s hygienic condition analysis revealed that dogs that were bathed regularly had lower tick infestation (13.3%) as compared to those which were not cleaned regularly (47.7%) (*p* >0.05). On the other hand, tick prevalence was also higher (56.3%) in the dogs with previous tick history than the dogs with no previous tick infestation history (4.7%) (*p* >0.05). Tick prevalence rate with respect to the host’s hygienic condition and previous exposure to tick bites were found to be statistically non-significant. Moreover, free-roaming dogs were found to be highly infested (47.7%) with ticks as compared to non-roaming dogs (14.7%) (*p* <0.05) (Table 5).

#### 3.2.4. Spatio-temporal Distribution of Tick Species

A total of 1150 ticks were collected from 300 dogs across the study area. Among the collected tick species, *Rh. sanguineus s.l.* ticks were abundant in district Malakand (42.7%) followed by Peshawar (42.5%), Swabi (41.5%), Mardan (39.5%), and Charsadda (40.2%), respectively. Whereas, *Rhipicephalus turanicus s.s.* ticks were found to be most prevalent with 32.9%, 29%, 28.2%, 26.9%, and 26.4% rates in districts Mardan, Charsadda, Peshawar, Swabi, and Malakand. Similarly, *Rh. haemaphysaloides* were richly distributed in district Charsadda (11.2%) followed by Swabi (10.4%), Mardan (10.1%), Peshawar (9.7%), and Malakand (8.8%). *Rhipicephalus microplus* was the least prevalent tick species, with prevalence rates of 4.4%, 2.4%, 2.2%, 0.9%, and 0.4% in Mardan, Swabi, Malakand, Charsadda, and Peshawar. Among the *Hyalomma* tick species, the distribution pattern of *Hy. dromedarii* was observed at rates of 16.7%, 16.5%, 15.4%, 15.2%, and 11% in districts Malakand, Swabi, Peshawar, Charsadda, and Mardan, while *Hy. excavatum* was the least abundant species (2.0%, 3.0%, 2.6%, 3.7%, and 3.7%) across the different districts of the study area. Overall, spatial distribution analysis showed that *Rh. sanguineus s.l.* (41.3%) was the predominant tick species followed by *Rh. turanicus s.s.* (28.7%), *Hy. dromedarii* (15%), *Rh. haemaphysaloides* (10%), *Hy. excavatum* (3%), and *Rh. microplus* (2%) respectively (Figure 2).

Temporal distribution of the collected tick species showed that the highest tick abundance was observed during the summer season, i.e., June (28.7%) followed by July (19.6%), May (16.8%), and August (15.0%), respectively. In the winter season, the highest number of ticks was collected during February (5.4%) as compared to January (3.9%). In spring, a large number of ticks were collected in April (8.9%), while the lowest was in March (6.5%). Based on different instars of ticks, month-wise distribution indicated that there was a gradual increase in the number of both adults and nymphs from January to June and then declined gradually toward August (Figure 3).

### 3.3. Potential Risk Factors for Tick Infestation

Univariable and multivariable logistic regression analyses were computed using the dataset to determine the potential risk factors of interest. The univariate regression analysis indicated both demographic (age, gender, and breed) and host management practices (acaricidal application, dog bathing, previous tick infestation history, and dog roaming range) were statistically significant (*p* < 0.05) risk factors associated with a tick infestation in the dog population. However, in multivariable logistic regression analysis, host gender, bathing practice, and dog roaming range were found to be statistically non-significant (*p* > 0.05), while all other variables such as age, breed, acaricides application, and previous tick infestation history were identified as potential determinants for tick infestation in the dog population (*p* < 0.05) (Table 6).

### 3.4. Molecular Attributes of Phylogenetic Markers of Collected Tick Species

The final *cox1* and *16S rRNA* amplicons’ sizes were ~621 bp and ~380 bp. All the *cox1* partial nucleotide sequences of collected tick species were AT-rich, as they reached 69.8% A+T for *Rh. turanicus s.s.*, 69.6% A+T for *Rh. sanguineus s.l.*, 69.2% A+T for *Hy. excavatum*, 68.4% A+T for *Hy. dromedarii*, 68.3% A+T for *Rh. haemaphysaloides*, and 67.9% A+T for *Rh. microplus*. Similarly, partial nucleotide sequences of the *16S rRNA* gene were AT-richer, as well, by 79% A+T for *Hy. dromedarii*, 78.9% A+T for *Rh. microplus*, 78.1% A+T for *Hy. excavatum*, 77% A+T for *Rh. turanicus s.s.*, 76.6% A+T for *Rh. sanguineus s.l.*, and 76.4% A+T for *Rh. haemaphysaloides* respectively (Appendix A).

Sequenced isolates of *cox1* and *16S rRNA* genes from the present study shared maximum similarities to other identical sequence isolates published in the NCBI GenBank database, mostly from neighboring Asian countries, i.e. China, Iran, and Turkey. Cytochrome c oxidase subunit 1 isolates of all tick species from the present study showed 100% identity with the same tick isolates published globally, except *cox1* isolates of *Rh. turanicus s.s.*, which shared 99.79% sequence similarity with subject sequences available in the NCBI GenBank (Appendix A).

Likewise, partial nucleotide sequences of the *16S rRNA* gene of the tick samples from the present study were found to be 100% similar to their identical tick species reported from China, Pakistan, Turkey, and Senegal respectively (Appendix A).

### 3.5. Phylogenetic Analysis: Neighbor-Joining

To infer the evolutionary relationship of collected tick species, *cox1* and *16S rRNA* partial nucleotide sequences-based phylogenetic trees were constructed using the Neighbor-Joining algorithm. In the *cox1* phylogram, *Psoroptis ovis* (OL913869) was used as an outgroup, whereas the *16S rRNA* phylogram was rooted with *Sarcoptes scabiei* (AB821000) to determine the direction of evolution of the reported tick species. Phylogenetic analyses of the collected tick species showed that *Hy. excavatum* ticks of this study clustered with similar isolates from India (MK736268, MK005261) and Iran (KX911989) with bootstrap support of 96. Likewise, *Hy. dromedarii* ticks grouped with identical specimens published from Iran (KT920181), Iraq (KM235696), Kenya (MT896151), and Saudi Arabia (MZ348812) with strong bootstrap support of 100. Ticks of the genus *Rhipicephalus*, commonly associated with ruminants but collected from dogs during this study, were also characterized and they shared evolutionary relatedness with similar isolates published from other agroecological zones of Pakistan and India. For instance, *Rh. haemaphysaloides* shared an evolutionary relationship with identical tick isolates from Pakistan (MT800317) and India (MW078974). Similarly, *Rh. microplus* ticks involved in this work clustered with identical tick sequences from Pakistan (MK462194) and India (KP792572, KX228541) with strong bootstrap support for both of the aforementioned ticks. The dog tick *Rh. turanicus s.s.*, clustered with previously reported isolates from Pakistan (MZ424825, MT800314) had a bootstrap value of 93, whereas the brown dog tick *Rh. sanguineus s.l.* from the present study shared an evolutionary relationship with the same isolates published from other countries, i.e., China (MG969507), Angola (MF425995), Brazil (KX383817), Colombia (KT906184), Thailand (MZ401443), and the Philippines (MZ726445), with bootstrap support of 69 (Figure 4).

The *cox1*-based molecular phylogenies of the collected tick species were further confirmed by inferring their evolutionary history through the *16S rRNA* phylogram. *Hyalomma excavatum* tick isolates of the present study projected on the same clade with Turkish (MT229183, KR870972, OL347856) and Indian (KP210042, KP210047) isolates of the same tick species. Meanwhile, *Hy. dromedarii* ticks of this study clustered with similar tick specimens from African countries, such as Senegal (KU130425), Algeria (OL672220), Kenya (MT895170), and Egypt (MF946464, KY945490) with a bootstrap value of 99. Similarly, *Rh. microplus* ticks clustered with reported identical isolates from Asia, including India (MF946459), China (KU664521), and Pakistan (MK495912) by bootstrap support of 96, whereas *Rh. haemaphysaloides* ticks grouped with the same isolates previously reported from Pakistan (MZ436881, MT799956) by considerable bootstrap support i.e. 90. The *Rh. turanicus s.s.* of this study shared a group with the same isolates available in the NCBI GenBank from Pakistan (MT99955) and Afghanistan (KY111474), with a statistical bootstrap value of 98. On the other hand, the brown dog tick *Rh. sanguineus s.l.* isolates shared a phylogenetic clade, with the same tick isolates infesting dogs inter-continentally, i.e. China (MG651947), Mexico (MH018820), Côte d’Ivoire (KX793745), Costa Rica (KT382449), and Thailand (KC170744) (Figure 5).

The *cox1* partial nucleotide sequences-based genetic divergence analysis showed that the isolate of *Hy. excavatum* from the study area represented 2% genetic divergence with specimens from Iran (KX911989) and India (MK736268, MK005261), whereas *Hy. dromedarii* showed 100% nucleotide identities with specimens from Iran (KT920181), Iraq (KM235696), Kenya (MT896151), and Saudi Arabia (MZ348812). On the other hand, *Rh. haemaphysaloides* showed no considerable divergence to Pakistani isolates (MT800317) and only 1% nucleotide divergence to Indian isolate (MW078974). *Rhipicephalus turanicus s.s.* exhibited 2.7% genetic divergence with previously reported isolates from Pakistan (MT800314, MZ424825), while *Rh. microplus* showed no genetic divergence with previously reported isolates from Pakistan (MK462194, KP792572, KX228541). Similarly, *16S rRNA* partial nucleotide sequences-based analysis indicated that *Hy. excavatum* exhibited 2.3% genetic divergence, whereas *Hy. dromedarii* showed no nucleotide divergence to the same species isolates published from Senegal (KU130425), Algeria (OL672220), Kenya (MT895170), and Egypt (KY946490, MF946464). *Rhipicephalus haemaphysaloides* showed 0.3% nucleotide divergence with previously reported Pakistani isolates (MZ436881 MT799956) and *Rh. turanicus s.s.* showed only 0.4% nucleotide divergence with published isolates from Pakistan (MT799955) and Afghanistan (KY111474). While *Rh. microplus* isolates from the study area showed 0.2% nucleotide divergence from isolates published from Pakistan (MK495912), India (MF946459), and China (KU664521).

### 3.6. Phylogenetic Analysis: Maximum Likelihood (Rh. sanguineus complex)

Cytochrome c oxidase subunit 1 and *16S rRNA* partial nucleotide sequences-based phylograms were inferred using the ML algorithm to demonstrate the evolutionary relatedness of *Rh. sanguineus s.l.* ticks of this study to the tropical lineage in an independent cluster. In *cox1*-based phylogram, *Rh. sanguineus s.l.* isolates of this study clustered with similar tick specimens from Brazil (MT010523, KX383820), Fiji (MK967893), Australia (MK967943), Thailand (MZ401443), Côte d’Ivoire (KX757914), Angola (MF425995), and Colombia (KT906184) in the tropical lineage (Figure 6). However, *Rh. sanguineus s.l.* isolates of this study were genetically diverged by 7.7–8% from the temperate lineage’s specimens of a brown dog tick. In comparison to, the ML-phylogram based on *16S rRNA* nucleotide sequences revealed that *Rh. sanguineus s.l.* under this study clustered with tropical lineage specimens from Mozambique (JX195173), Brazil (GU553075), Colombia (GU553076), Cuba (JX997389), Angola (MF425981), Kenya (KU746973), South Africa (JX195174), Egypt (KY413782, KY413777), and Argentina (JX206980). The tropical lineage included *Rh. guilhoni*, *Rh. sulcatus*, and *Rh. afranicus* ticks, as well. Based on the *16S rRNA* phylogram, *Rh. sanguineus s.l.*, under this study, genetically diverged by 9.7–10% from the temperate lineage’s specimens (Figure 7).

## 4. Discussion

In Pakistan, the environmental conditions are conducive to tick reproduction and development. Although, several studies have identified ticks from diverse hosts across different geographical locations in Pakistan [23,25,26,36,37,38,39]. Only one study had cataloged the tick species morpho-taxonomically infesting dogs thus far, from the Baluchistan province of Pakistan [17]. To the best of our knowledge, this is the first detailed report on the epidemiology, molecular characterization, and genetic diversity of hard ticks infesting dogs (with a special focus on the brown dog tick *Rh. sanguineus s.l.*) across Khyber Pakhtunkhwa, Pakistan.

In the current study, the overall prevalence of tick infestation was 61% in dogs, while across different districts, a higher prevalence of ticks was observed in district Mardan (14.7%) followed by Peshawar (13%), Swabi (12%), Charsadda (11%), and Malakand (10.3%). In the last three decades, several studies have attempted to report the prevalence rates of tick infestation, ranging from 6.9% to 86.5% in livestock across different geographical localities of the country [16,23,25,26,37,38,39,40,41,42,43,44,45,46,47,48]. Our results also fall within the same range and support these reports. Higher tick infestation was observed in the dog population during the present study as compared to the previously published reports from India, Nigeria, Pakistan (Lahore), and Iran [49,50,51]. Their findings include tick infestation at the prevalence rate of 45%, 55.3%, 52.3%, and 53%, respectively. On contrary, a higher prevalence rate (71.2% and 96.0%) of ticks infesting dogs was observed in the Ilorin and Maiduguri region of Nigeria [52]. These differences in the tick prevalence rate may be due to different geographic and eco-climatic conditions as well as different sample sizes.

Risk factors associated with tick infestation are highly uneven among scientific studies. Regarding host demographic attributes, female dogs were found to be more often infested with ticks than male counterparts. The possible explanation in support of these observations is that female dogs have frequent hormonal fluctuations and disturbed immune systems during the gestation/lactation period and have a sedentary lifestyle while nursing their puppies, which allows questing ticks to infest them and thus carry more ticks than male dogs [53,54,55]. However, further studies are needed to characterize this phenomenon. The present study also revealed that the young dogs (puppies and juveniles) were highly infested as compared to adult dogs. These findings corroborate the previous studies conducted in India, Iran, and the Mexico–USA border region [56,57,58,59]. The rationales for this phenomenon are: that puppies and young dogs have an immature immune system that is not fully developed/functional, while adults have repeated exposure to tick infestation, so their immune system more effectively resists tick infestation as compared to juveniles [52,59,60].

Similarly, among the management factors, a preventive measure such as acaricidal application can reduce the tick infestation rate. In our results, tick infestation in dogs was significantly associated with acaricidal use. For instance, low tick infestation was observed in the dogs treated regularly with acaricides as compared to the irregular and non-treated dogs. It has been demonstrated that the effect of acaricides is distributed uniformly on the skin surface of dogs, indicating that the increasing distance from the application site causes no changes in the effect of the active ingredients of acaricides [61], and therefore successfully detach ticks from the dog’s body.

The season for the onset of tick activity is of great concern, particularly for veterinarians and pet owners, to adopt the best precautionary measures for controlling tick infestation and tick-borne pathogens’ transmission. In the present study, the seasonal distribution of ticks revealed a biphasic pattern of tick activity, i.e., the summer phase, when ticks are highly active, and the winter phase, when ticks show minimal activity, which is in parallel with literature published from central Europe [54,55,62] that reported a similar biphasic pattern of tick infestation in dogs. Moreover, during the present study, a gradual increase was observed in tick abundance from January through April to May, reaching the peak in June, and then starting a gradual decline in July and August. This may be assumed that a gradual temperature rise toward the optimum can enhance tick questing behavior much more efficiently than it would be during an overall winter and results in high tick infestation of the host [55,62,63]. In addition, the summer season (May–June), with optimum temperature and relative humidity, provides suitable environmental conditions for tick growth and reproduction. However, increased precipitation due to monsoon (July–August) may cause a decrease in the tick infestation [18,27,64].

The morpho-molecular-based identification of ticks represented two genera, *Hyalomma* and *Rhipicephalus*, and six species infesting dogs, including *Hy. excavatum, Hy. dromedarii*, *Rh. microplus*, *Rh. haemaphysaloides, Rh. sanguineus s.l.*, and *Rh. turanicus s.s.* Our findings corroborate a previous study carried out in the Baluchistan province of Pakistan that reported a similar pattern of tick species composition in dogs [17]; however, our results did not comply with the presence of any *Haemaphysalis* species in dogs during the study period. Among the identified tick species, *Rh. sanguineus s.l.* was predominant, followed by *Rh. turanicus s.s.*, *Hy. dromedarii*, *Rh. haemaphysaloides*, *Hy. excavatum*, and *Rh. microplus*, respectively. In context to the cosmopolitan distribution of *Rh. sanguineus s.l.* ticks worldwide [8,9,65], the present study molecularly confirmed the presence of this tick species on dogs for the first time in Pakistan. Primarily, dogs are the preferred hosts of this tick species [7,13] that are in agreement with our findings, with occasional infestations on rodents, birds, and humans [10].

Morphologically, there is a global conflict regarding the accurate identification between the members of the *Rh. sanguineus* complex, particularly the closely similar ones. Despite the description of the *Rh. sanguineus s.s.* neotype [12], continuous misidentification of most of the members of the complex still occurs [66]. Although there are some morpho-taxonomic records of *Rh. sanguineus s.l.* from Pakistan [17,26,67], to date no molecular evidence has been provided to solve and confirm the status of the actual species and its lineage from Pakistan. Molecularly, earlier works have revealed that a minimum of three distinguished mitochondrial lineages of *Rh. sanguineus s.l.* do exist, namely, tropical lineage (tropical areas), temperate lineage (cold regions) [68,69,70], and southeastern Europe lineage (north Africa and south/southeastern Europe) [11]. Our findings confirmed that *Rh. sanguineus s.l.* ticks from this study belong to the tropical lineage, which included the same tick specimens from Brazil, Fiji, Australia, Thailand, Panama, Angola, Colombia, Egypt, South Africa, Cuba, Kenya, Argentina, Mozambique, and the Ivory coast. The *Rh. sanguineus s.l.* specimens of the tropical lineage are hypothesized to be referred to as *Rh. linnaei* based on an original description from Egypt and distribution in Australia [71]. However, such a hypothesis is still under review; if confirmed, then *Rh. sanguineus s.l.* of this study should be referred to as *Rh. linnaei*.

*Hyalomma excavatum* ticks prefer bovine, ovine, and equine as hosts with possible infestations in humans [7,17,72,73,74]; however, they have been recorded attacking dogs in the current study, presumably increasing their host range to include canines as a new association. Similarly, *Hy. dromedarii* is considered an Afrotropical, Oriental, and Palearctic species, distributed throughout the Middle East and Central Asia, India, the Arabian Peninsula, and several parts of Africa [7,75], although several studies are confirming the occurrence of this species in Pakistan and on the Iran–Pakistan borders [76,77,78]. No single study presented the molecular aspects of this species collected from dogs in the country to date. We also present the first detailed molecular study on both of these species of ticks infesting canine populations across the study area. Several studies reported hyenas, dogs, ostriches, reptiles, and humans as rare hosts for *Hy. dromedarii* [7,79,80,81]; our findings are approved as one of the rare associations between this tick species and dogs.

Due to the high genetic variability in *Rh. sanguineus s.l.* ticks, ML phylogenetic trees were constructed based on both *cox1* and *16S rRNA* partial nucleotide sequences, to uncover the better resolution of the lineage belonging to the specimens under study. Based on the *cox1* isolates, *Rh. sanguineus s.l.* of this study belongs to the tropical lineage, as it clustered with similar isolates from Brazil, Fiji, Australia, Thailand, Panama, Angola, Colombia, and the Ivory Coast. Likewise, it clustered in the same tropical lineage based on *16 rRNA* gene sequences with isolates from Argentina, Egypt, South Africa, Kenya, Angola, Cuba, Colombia, Brazil, and Mozambique along with *Rh. afranicus, Rh. sulcatus*, and *Rh. guilhoni*. Genetic divergence analysis indicated that *cox1* and *16S rRNA* isolates exhibit diversity from the subject sequences of similar tick species by 9.6–10% and 7.7–8%. In context to this high genetic divergence, previous studies have confirmed it between different lineages of *Rh. sanguineus s.l.* [66,68]. Such considerable genetic diversity could be due to many factors, i.e., genetically, the founder effect could be a reason for such a high divergence with mutations as a result of individual/geographical isolation away from the original population, presenting different gene pools and compositions [82], which is in agreement with the cosmopolitan distribution of *Rh. sanguineus s.l.* ticks.

## 5. Conclusions

Several studies were carried out regarding tick surveillance and identification based on morpho-taxonomic criteria. It becomes inaccurate and sometimes misleading with reliance on morphological characteristics alone, which can result in misidentification by those with limited expertise. Molecular tools linked to the proper identification of biological specimens have appeared to avoid these difficulties in the correct identification and characterization of tick fauna and provide better resolution about tick genetics and evolutionary relationships. In the present study, we performed a molecular investigation of ticks infesting dogs in Pakistan based on *cox1* and *16S rRNA* genes. This study explored new associations between tick–host (*Hy. excavatum* and *Hy. dromedarii* to dog hosts) across the study area. The present study also confirmed that the *Rh. sanguineus s.l.* tick from Pakistan falls in a tropical lineage. Epidemiological profiles of the ixodid ticks infesting canine hosts were also established. Further molecular works need to be performed throughout Pakistan to present a more detailed map of tick distribution with information about dog-host associations.

## Figures and Tables

**Figure 1 pathogens-12-00098-f001:**
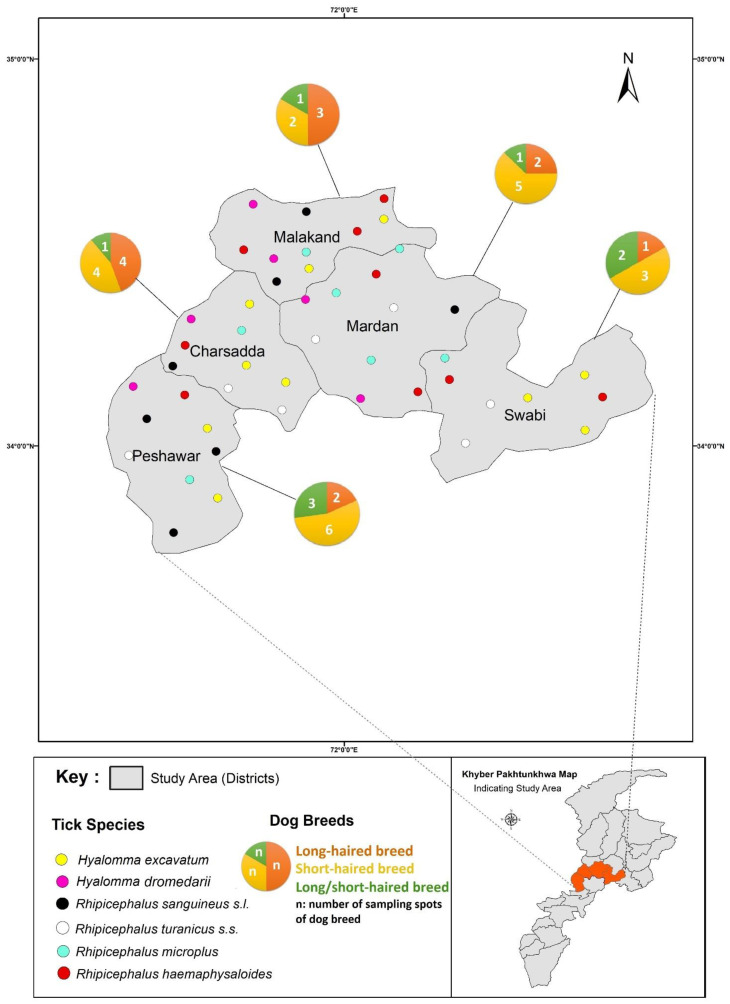
Map of Khyber Pakhtunkhwa Pakistan, representing tick sampling spots/host distribution area.

**Figure 2 pathogens-12-00098-f002:**
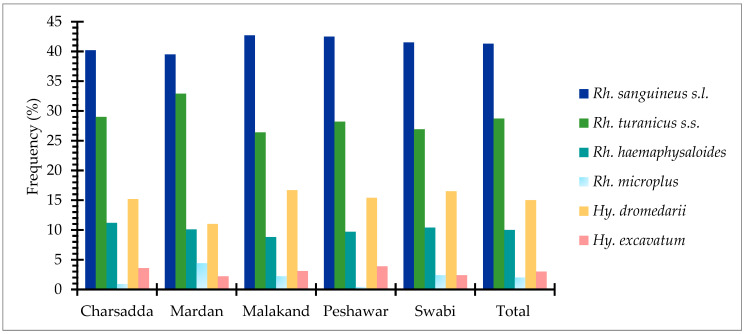
Spatial distribution of ixodid tick species across the study area.

**Figure 3 pathogens-12-00098-f003:**
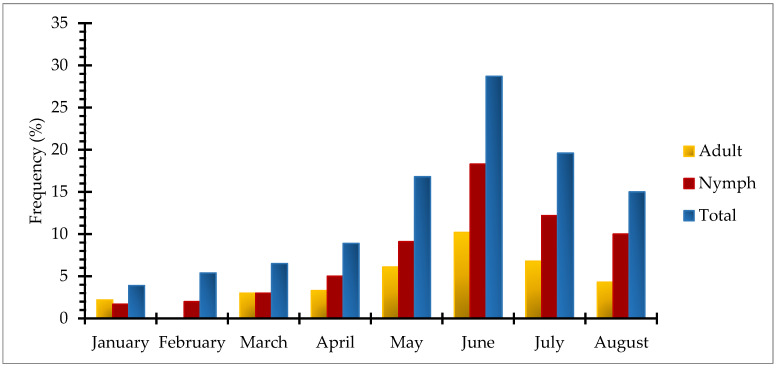
Temporal distribution of ixodid tick species infesting dogs across the study area.

**Figure 4 pathogens-12-00098-f004:**
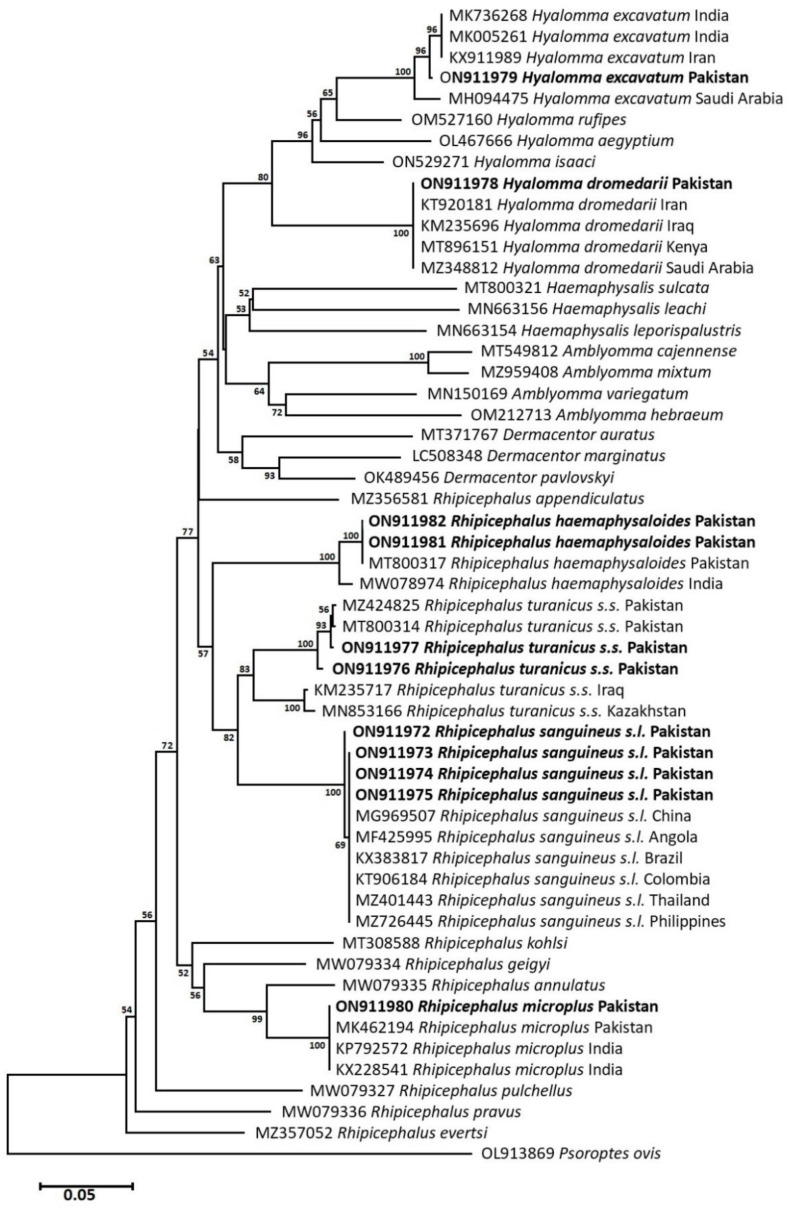
Neighbor-Joining phylogram inferred from *cox1* partial nucleotide sequences of ixodid ticks from this study (Bold) and other identical isolates available in GenBank. The tree was constructed and analyzed by the NJ algorithm using a General Time Reversible model with a 1000 bootstrap value. *Psoroptes ovis* was used as an outgroup.

**Figure 5 pathogens-12-00098-f005:**
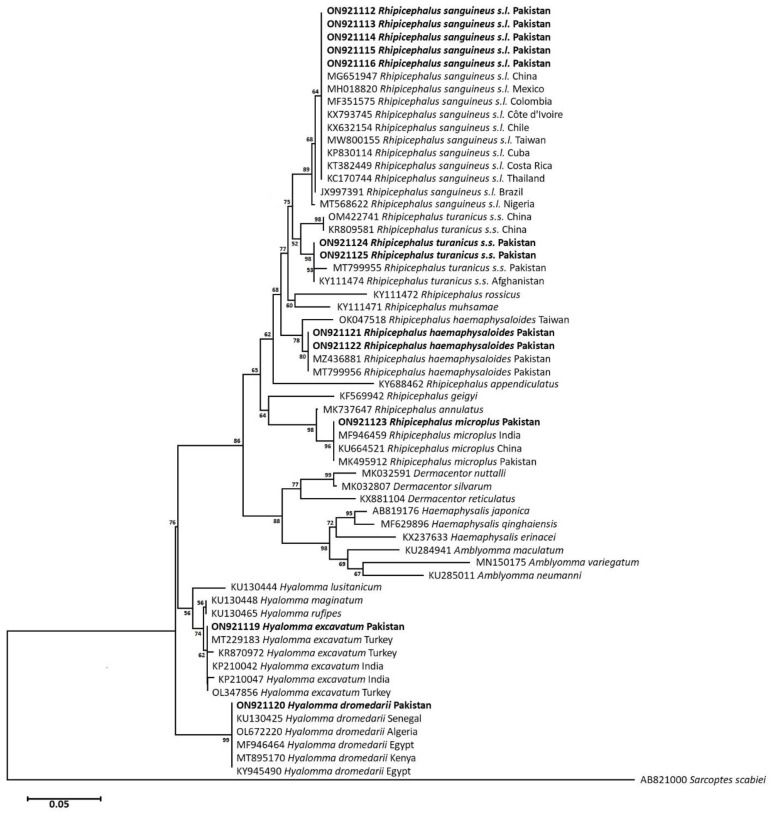
Neighbor-Joining phylogram inferred from *16S rRNA* gene partial sequences of ixodid ticks from this study (Bold) and other identical tick isolates from the NCBI GenBank. The tree was constructed and analyzed by the NJ algorithm using the Tamura 3-parameter model with a 1000 bootstrap value. *Sarcoptes scabiei* was used as an outgroup.

**Figure 6 pathogens-12-00098-f006:**
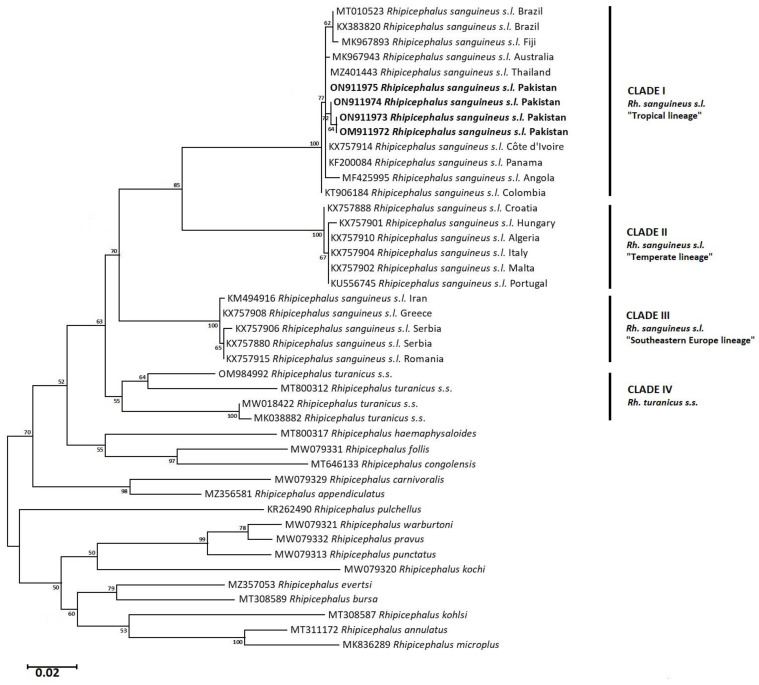
Maximum likelihood (ML) tree based on *cox1* partial nucleotide sequences of *Rhipicephalus sanguineus s.l.* isolates from this study (in bold) and members of the genus *Rhipicephalus* published in NCBI GenBank. The branch numbers represent bootstrap support (1000 replicates). The scale bar shows phylogenetic distance. The tree was constructed using ML algorithm with a general time reversible model.

**Figure 7 pathogens-12-00098-f007:**
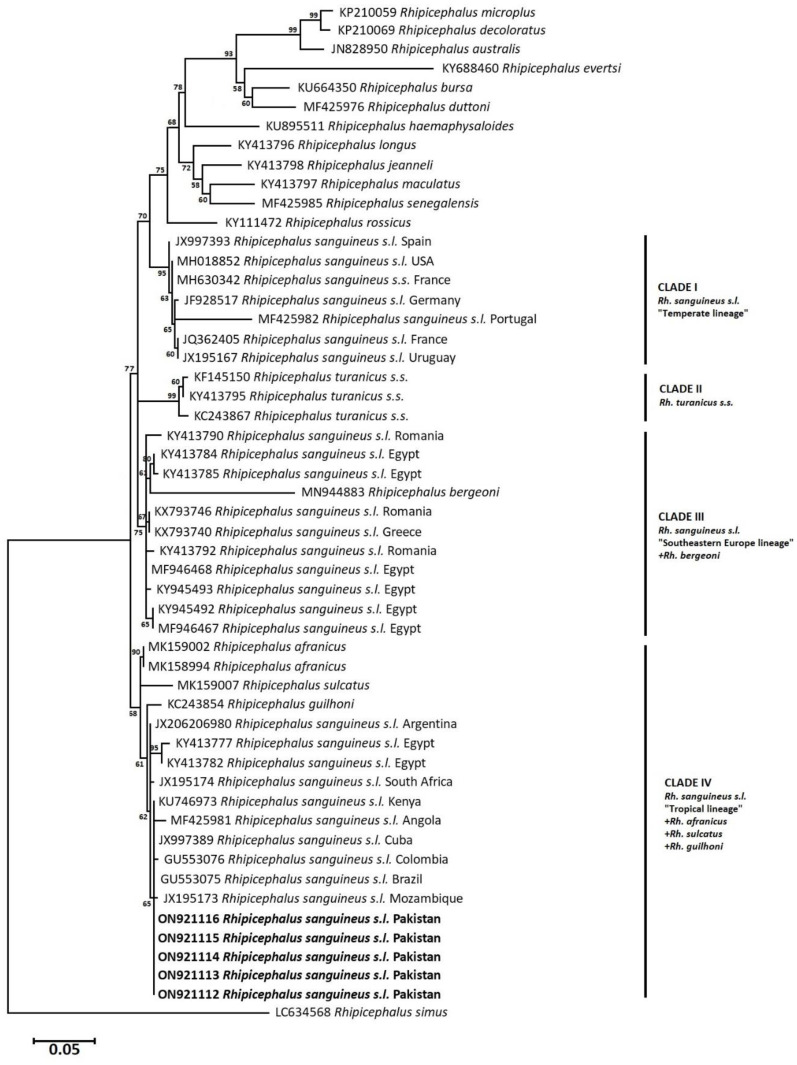
Maximum likelihood (ML) tree based on *16S rRNA* partial nucleotide sequences of *Rhipicephalus sanguineus s.l.* isolates from this study (in bold) and members of the genus *Rhipicephalus* from NCBI GenBank. The branch numbers represent bootstrap support (1000 replicates). The scale bar shows phylogenetic distance. The tree was constructed using ML algorithm with the Tamura 3-parameter model.

**Table 1 pathogens-12-00098-t001:** List of primer sets used for the amplification of target genes of ticks.

Organism	Primer Name	Primer Sequence (5′–3′)	Target Gene	Product Size (bp)	Reference
Ticks	LCOI490	GGTCAACAAATCATAAAGATATTG	*cox1*	~710	[31]
	HCO2198	TAAACTTCAGGGTGACCAAAAAATCA			
Ticks	16S+1	CTGCTCAATGATTTTTTAAATTGCTGTGG	*16S rRNA*	~460	[32]
	16S-1	CCGGTCTGAACTCAGATCAAGT			

**Table 2 pathogens-12-00098-t002:** Demographic profile of the host population from the study area.

Demographic Variable	Category	Study Area (Districts)					Total *n* (%)
		Charsadda *n* (%)	Mardan *n* (%)	Malakand *n* (%)	Peshawar *n* (%)	Swabi *n* (%)	
Age (year)	Puppy (<1)	21 (12.5)	31 (25.2)	24 (19.5)	28 (22.8)	19 (15.4)	123 (41)
	Juvenile (1–3)	17 (17.2)	23 (23.2)	19 (19.2)	25 (25.3)	15 (15.2)	99 (33)
	Adult (>3)	14 (18.0)	12 (15.4)	19 (24.4)	18 (23.0)	15 (19.2)	78 (26)
Gender	Female	38 (20.7)	28 (15.2)	39 (21.2)	40 (21.7)	39 (21.2)	184 (61)
	Male	22 (19)	23 (19.8)	24 (20.7)	26 (22.4)	21 (18.1)	116 (38.6)
Breed	Short-haired	33 (18.1)	36 (11.5)	38 (20.9)	41 (22.5)	34 (18.7)	182 (60.6)
	Long-haired	23 (19.5)	22 (18.6)	25 (21.2)	27 (22.9)	21 (17.8)	118 (39.3)
Dog roaming range	Free-roaming	41 (20.4)	31 (15.4)	42(20.8)	44 (22.0)	43 (21.4)	201 (67.0)
	Non-roaming	19 (19.1)	20 (20.2)	20 (20.2)	22 (22.4)	18 (18.1)	99 (33.0)

**Table 3 pathogens-12-00098-t003:** Total and district-wise prevalence of ticks in dog population across the study area.

Host	Study Area (Districts)										Total Prevalence *n* (%)
Dog	**Charsadda**		**Mardan**		**Malakand**		**Peshawar**		**Swabi**		
	NHE	NHTI (%)	NHE	NHTI (%)	NHE	NHTI (%)	NHE	NHTI (%)	NHE	NHTI (%)	
	52	33 (11.0)	66	44 (14.7)	62	31 (10.3)	71	39 (13.0)	49	36 (12.0)	183/300 (61%)

NHE: number of hosts examined for ticks, NHTI: number of hosts tick-infested (prevalence).

**Table 4 pathogens-12-00098-t004:** Ixodid ticks’ prevalence with respect to host demography.

Host Demographic Variable	Category	Examined Host *n* (%)	Tick-Infested Host *n* (%)	95% CI	Chi (*χ^2^*) Statistic	*p*-value
Age (year)	Puppy (<1)	123 (41.0)	104 (34.7)	29.3–40.1	12.210	0.001
	Juvenile (1–3)	99 (33.0)	54 (18.0)	13.6–22.3		
	Adult (>3)	78 (26.0)	25 (8.3)	5.1–11.4		
Gender	Female	184 (61.0)	127 (42.3)	36.7–47.9	6.115	0.01
	Male	116 (38.6)	56 (18.7)	14.3–22.1		
Breed	Short-haired	182 (60.6)	126 (42.0)	36.4–47.5	4.232	0.05
	Long-haired	118 (39.3)	57 (19.0)	14.5–23.4		

**Table 5 pathogens-12-00098-t005:** Ixodid ticks’ prevalence with respect to the host management practices.

Host Management Variable	Category	Examined Host *n* (%)	Tick-Infested Host *n* (%)	95% CI	Chi (*χ^2^*) Statistic	*p*-value
Acaricides	No use	168 (56.0)	130 (43.3)	37.7–48.9	17.989	0.001
	Irregular use	85 (28.3)	38 (12.7)	8.9–16.5		
	Regular use	47 (15.7)	15 (5.0)	2.5–7.5		
Dog bathing	No	214 (71.3)	143 (47.7)	42.0–53.3	2.391	0.10
	Yes	86 (28.7)	40 (13.3)	9.5–17.1		
Previous tick infestation	Yes	238 (79.3)	169 (56.3)	50.7–61.9	5.110	0.10
	No	62 (20.7)	14 (4.7)	2.3–7.1		
Dog roaming range	Free-roaming	201 (67.0)	143 (47.7)	39.1–50.3	6.351	0.01
	Non-roaming	99 (33.0)	40 (14.7)	10.7–18.7		

**Table 6 pathogens-12-00098-t006:** Potential risk factors facilitating tick infestation of the host population.

Demographic/Host Management Associated Variable	Tick-Infested Host *n* (%)	Univariate Logistic Regression Analysis					Multivariate Logistic Regression Analysis				
		β	OR	(95% CI)		*p*-value	β	OR	(95% CI)		*p*-value
				LL	UL				LL	UL	
Age (year)											
Puppy (< 1)	104 (34.7)	1.78	3.41	2.45	4.83	0.001	1.66	4.49	2.23	9.18	0.001
Juvenile (1–3)	54 (18.0)										
Adult (> 3)	25 (8.3)										
Gender											
Female	127 (42.3)	1.25	2.38	1.48	3.87	0.001	−0.66	0.26	0.05	1.10	0.08
Male	56 (18.7)										
Breed											
Short-haired	126 (42.0)	1.43	2.41	1.49	3.90	0.001	−0.09	0.08	0.01	0.48	0.01
Long-haired	57 (19.0)										
Acaricides											
No use	130 (43.3)	1.88	2.97	2.12	4.24	0.001	1.02	2.36	1.32	4.41	0.004
Irregular	38 (12.7)										
Regular	15 (5.0)										
Dog bathing											
Yes	143 (47.7)	1.14	2.07	1.24	3.48	0.005	0.69	0.90	0.37	2.13	0.11
No	40 (13.3)										
Previous tick infestation											
Yes	169 (56.3)	3.40	8.69	4.61	17.32	0.001	1.01	3.15	1.13	9.19	0.03
No	14 (4.7)										
Dog roaming range											
Free-roaming	143 (47.7)	1.92	3.63	2.21	6.06	0.001	1.27	3.7	0.61	33.6	0.18
Non-roaming	40 (14.7)										

β: regression coefficient, OR: odds ratio, CI: confidence interval at 95%, LL: lower limit, UL: upper limit.

## Data Availability

The dataset and related information have been included in this manuscript.

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
