# Peer review of "Ticks Infesting Dogs in Khyber Pakhtunkhwa, Pakistan: Detailed Epidemiological and Molecular Report"

_pathogens, 2023, doi:10.3390/pathogens12010098_

Round 1

Reviewer 1 Report

To the Authors:

“Ticks infesting dogs in Khyber Pakhtunkhwa, Pakistan: First epidemiological and molecular report” is presented for review by Jehan Zeb et al. The authors present findings that include the presence of known and previously unknown species of ticks on dogs. They also present phylogenetic analyses of genetic data on a subset of each sample. The manuscript adds information known about tick abundance and species distribution of ticks found on domestic dogs in Pakistan. It is evident that this manuscript represents a lot of work, but the manuscript suffers somewhat from missing or insufficient methods, copy-paste errors, orphaned subheaders, and an overall lack of clarity. 

Specific areas of improvement are listed below:

ABSTRACT

Lines 38-40: font size is different

Line 45: font size is different

Line 45: This is the first mention of long-hair breeds, but you mention medium-hair breeds in the results, but no long-hair breeds. Please be consistent or clarify why medium-hair breeds are mentioned if they are not in the methods (and why long-hair breeds are not mentioned in the results).

INTRODUCTION

Lines 53-55: Several important tick species are found on human and nonhuman animals in areas other than tropical and sub-tropical regions. Rhipicephalus sanguineus s.l. occurs in both desert and neotropical regions of North America and has been responsible for outbreaks of Rocky Mountain Spotted Fever. 

Lines 62-71: Not sure why this paragraph is here. Environmental factors are not tested in your paper. 

Line 68: “behaviour” should be plural.

Line 103: I recommend adding “mitochondrial” in front of this first 16S to distinguish it from bacterial 16S once. After specifying this once, "16S rRNA" as short-hand is sufficient.

Line 110: Here you specify the randomized sampling of owned and stray dogs, yet in the methods, nothing is mentioned about the stray dogs. See Methods for further comments.

MATERIALS AND METHODS

Figure 1: What are “mixed” hair breeds? This is not defined in the text.

Study design and tick sampling strategy

Lines 141-43: “Dogs of willing pet owners were included in this study”. In Line 110, you mentioned that both stray dogs and pet dogs were used. Here is where more information is needed. How many dogs were strays? how many were pets? How did you obtain information on the strays that could be used for comparison to the pets? Were these mark/recapture strays? Or are you referring to dogs that are considered pets, but have a roaming range that is not in a confined area (e.g. yard or home)? This needs clarification.

Lines 147-148: You mention two types of breeds: long hair and short hair, yet in the results, you mention medium hair (line 249) and also in Table 5. Please clarify or correct.

Line 152: what is a mouth gag? is that different from a muzzle?

Line 156: “shifted” change to “shipped”

Tick genomic DNA extraction and gene amplification: What was the purpose of using only 120 ticks (out of 1150) for molecular assay? If you wanted to confirm identification by assay for each location, wouldn’t you want to test a higher proportion by location, not 10% across the whole collection? Why did you assay 20 samples per taxon?  

Line 168; Table 1: References for the primers are listed as the authors. Unless the sequences presented are incorrect, these sequences are the same as those from the original source, Folmer et al 1994 and Black& Piesman 1994, respectively. If you had modified the primers in some way, perhaps citing yourself is acceptable, but as presented it is not.

Amplicon purification and DNA sequencing

 Line 176: “MEGA VII” should be “MEGA7” (https://pubmed.ncbi.nlm.nih.gov/27004904/). The version after that was MEGA X, with a space and the Roman numeral, but the versions before and after were “MEGA7” and “MEGA11”, respectively (https://www.megasoftware.net/citations).

It looks like you sequenced 11 tick samples across taxa for the cytochrome oxidase subunit I and 12 for mitochondrial 16S rRNA.You have not specified from where these samples were collected, which is unfortunate since you had some information on district-specific tick data. 

Phylogenetic analysis

Line 184 (see comment above)

Lines 182-189: In the Results section, you present 4 phylogenetic trees. The first two are Neighbor-joining trees that include all the sequences for the COI and 16S rRNA amplicons, respectively, but these have different outgroups (which you do not specify here). The outgroups in the results are not the same as those listed in one of the Figures (Figure 6 states that the outgroup for 16S is Psoroptes ovis, but the text in the Results section states that it is Sarcoptes scabiei). Please check for correctness.

The second pair of phylogenetic trees is analyzed with a Maximum Likelihood analysis, but the approach is slightly different (you did not include outgroups and it seems to be focused exclusively on the genus Rhipicephalus). Please explain why you did not use an outgroup here, but. you used one in the aforementioned tree. (Note that without an outgroup, you can discuss the clustering of sequences (relatedness within a group), but not the direction of evolution).

Statistical analysis

Logistic regression is used, but not specified. What analysis/analyses? In the results, p-values are presented without any associated test statistics. Yes, the presence of confidence intervals is helpful, but if we don’t know from what analyses they were derived, it doesn’t mean anything. Are these also from the regression or some other analysis (t-test? chi-square? other? if it is a regression, do you have a correlation coefficient)? I suspect some explanatory paragraph was missed during the copy-paste/editing process, but it does need to be addressed. 

You used odds ratios in some of your results yet you do not discuss this here or in your results.

RESULTS

Host-based demographic Characteristics <--why is "Characteristics" capitalized"?

Table 2: “List of reference primer sets used for the amplification of target genes of ticks” The title of this table is the same as Table 1. It is also cut off with headers on page 5 and the rest of the table on page 6. It is difficult to evaluate this table. I don’t know whether the numbers and percentages refer to the proportion of puppies of total dogs by district or the number/percentage of each subcategory with at least one tick by each district. As much of the data seems to be either unexplained or redundant in other figures/tables, perhaps add the data from Figure 2 into this table and remove Figure 2.

Overall district

Prevalence = # dogs with at least one tick, abundance=# ticks per dog? I assume the prevalence, in this case, is the former, but it is not clear since you haven’t defined this anywhere. I assume this means that 183 of 300 dogs had at least one tick, but no information about co-infestation by species, number of ticks per dog(abundance), or number of species per dog characteristic is presented.  

Figure 2: Could be included in Table 2 (after it is fixed) or if presented as is, please add the N to the figure.

Overall and spatio-temporal distribution of tick species 

Figure 3: Please add N to the figure. Comment: Good of you for trying to use a more colorblind-friendly palette, but you might want to check on "Colorbrewer" (https://colorbrewer2.org) to make sure the colors are not too similar.

Temporal distribution: These were the data from one year, so it does not represent variation from year to year, but it is suggestive of seasonal trends.

Line 242: Another orphan subheading. “Host demographic-based prevalence of tick infestation in dog populations” should be moved to the next page.

Lines 243-52: p-values are presented, but for what statistical analysis? CI is not an analysis.

Management factors-based prevalence of tick infestation in dogs 

Lines 255-264: Same comment for lines 243-52. The p-values are presented for what statistical analysis? There is a p-value missing for stray versus pet dogs. 

Line 255: “Environmental factors” which are what? It seems that you use “environmental” to be synonymous with “Management”, as listed in your subheading. Perhaps a copy-paste error?

Risk factors for the tick infestation 

Line 272-275: Font size is different

Table 5. Risk factors of tick infestations of dog hosts.: An odd ratio is binary (yes or no, sick or not, dead or alive, infested or not, etc.). If you do have an odds ratio for multiple categories you need an OR for each pairwise comparison. You need to clarify what you are presenting.

Line 277: Subheading “Molecular characteristics” This subheading should be moved to the next page. It looks like it was part of the “risk factors” table.

Line 278: “COICOI” should be “COI”

Phylogenetic analyses

Because there are two types of analyses presented here, you might want to have two subheadings: Phylogenetic analysis: Neighbor-joining and Phylogenetic analysis: Maximum Likelihood. It makes it easier to keep track of which analysis is being presented. 

Figures 5-8: General comments:

It might be helpful to include the district(s) from which the Pakistani R.s.s.l. sequences were derived. 

  The resolution is not great and some bootstrap values were difficult to read. Please make the bootstrap font size a bit larger (you can do this in MEGA before exporting the figure).

What was the actual fragment size after trimming? This should be in the figure legend and is provided by MEGA7 in the caption.

Figure 8: Egyptian specimens cluster with Clade III (the “south-eastern Europe lineage”) plus with another species. This broad clustering of Rssl plus three other species also occurred in Clade IV. If these were rooted in a sister taxon (e.g. Dermacentor) would these relationships be better resolved?

Line 320: “The 16S rRNA phylogenetic tree…” add “Neighbor-joining” before “phylogenetic” to distinguish this from the maximum likelihood analysis. 

Line 321 and Figure 6: Outgroups are not the same in the Figure legend as in the text or figure itself. Please correct.

Line 358: Suggested subheading: “Phylogenetic analysis - Maximum Likelihood”

Line 358: Not sure if “pertinence” is the right word. Perhaps “Rssl sequences from both COI and 16S rRNA amplicons from this study (Pakistan) cluster with the tropical lineage of Rssl (previously described by…)”.

As an aside, COI is the gold standard for barcoding but can be highly variable even within species (particularly in ticks). Thus, the use of mitochondrial 16S in conjunction with COI provides more robust support for your conclusions. Another point: phylogenies of mitochondrial sequences can also be misleading or biased, so ideally inclusion of a nuclear gene or one not under the same selection pressure is also preferred.

DISCUSSION

Lines 385-387: Period after “places of Pakistan [20, 22,28,29, 30,31]”. Then “One study compiled (catalogued?) tick species infesting dogs …, However, this study was …” 

Line 397: “We observed higher infestation levels (61%) on dogs from Pakistan than previously observed …”

**Note that your observations were for one year and do not reflect variability over time**

Line 405: “Risk factors associated with tick infestation are highly uneven between studies.” Perhaps “variable” in place of “uneven”? Is the unevenness referring to your studies or studies of ticks on dogs in general? 

Line 406-407: “female dogs were found with high(er?) tick infestation than male dogs, although no statistical difference was found”, but in univariate regression it was significant. This suggests that there are other factors contributing, but you do not address this. 

Line 407-416: You present two speculations here. The first is about females being more likely to get infested because they are taking care of puppies, but were all the females active mothers? This is not clear. The second was that puppies are in closer proximity to the ground and therefore have a higher likelihood of infestation than adults. This doesn’t make sense. They all have to walk on the ground and likely cover the same habitat. What you could have done (if there were enough data) was compare females that were with puppies versus females that were not with puppies and see if there was any correlation. 

Line 418-419: Reword this sentence. 

Line 424-5: Is it specific to the type of acaricide, or any use versus none? Was it known what type was used? (oral, topical, bath, spray, powder, etc). The frequency of treatment somewhere was mentioned as irregular or regular. How were those analyzed? 

Line 434: “gradual decline in July/August” —in some temperate species this corresponds to the decline of adults and the increase in nymphs, but not observed here. You suggest that the monsoon season may have contributed here. Line 439: High humidity is not the problem for ticks—they thrive in high humidity. “Increase in precipitation” makes sense based on the context of your sentence.

Line 455-46, “…continuous misidentification of most of the members of the complex is still on the rise” reword to “…complex still occurs”

Line 459: “woks” should be “works” 

Paragraph Lines 470-481 need some restructuring for flow:

Start with “Here we describe… a new canine association for Hy. excavatum. and H. dromedarii.” Hy excavatum ….” (Since you go into this at great lengths for Hy. dromedarii, you should probably also include geographic distribution for Hy. excavatum)

Next, describe Hy. dromedarii (hosts and geographic distribution: “Several studies reported hyenas, dogs, ostriches….” 

End with “We also present the first detailed molecular study on both of these species of ticks…”

Line 489: Note: Some of these Egyptian sequences clustered with the south-eastern Europe lineage. Can you discuss this?

Line 497-500: Be careful not to speculate too much about divergence—you can’t say much about evolutionary distance unless you root the tree.

Conclusion:

Line 502-507: Stop after “morphological identification”. Replace “however, it becomes inaccurate and sometimes misidentified without…” with “Reliance on morphological characteristics alone can result in misidentification by those with limited…”

Supplemental information:

Supplementary Tables 1 and 2: According to your phylogenetic tree and on Genbank the country is listed as “Pakistan” for these accession numbers, but in both tables, Rh.s.l. were listed as from China. You need to fix this.

Author Response

Comments and Suggestions for Authors

Reviewer 1st

To the Authors:

“Ticks infesting dogs in Khyber Pakhtunkhwa, Pakistan: First epidemiological and molecular report” is presented for review by Jehan Zeb et al. The authors present findings that include the presence of known and previously unknown species of ticks on dogs. They also present phylogenetic analyses of genetic data on a subset of each sample. The manuscript adds information known about tick abundance and species distribution of ticks found on domestic dogs in Pakistan. It is evident that this manuscript represents a lot of work, but the manuscript suffers somewhat from missing or insufficient methods, copy-paste errors, orphaned subheaders, and an overall lack of clarity. 

Specific areas of improvement are listed below:

ABSTRACT

Lines 38-40: font size is different

Response: The font size changed.

Line 45: font size is different

Response: Changed accordingly

Line 45: This is the first mention of long-hair breeds, but you mention medium-hair breeds in the results, but no long-hair breeds. Please be consistent or clarify why medium-hair breeds are mentioned if they are not in the methods (and why long-hair breeds are not mentioned in the results).

Response: It is long-hair breeds, not medium-hair breeds. It is a typo and the mentioned name was rectified accordingly.

INTRODUCTION

Lines 53-55: Several important tick species are found on human and nonhuman animals in areas other than tropical and sub-tropical regions. Rhipicephalus sanguineus s.l. occurs in both desert and neotropical regions of North America and has been responsible for outbreaks of Rocky Mountain Spotted Fever. 

The sentence is corrected to overcome the mentioned deficiency.

Lines 62-71: Not sure why this paragraph is here. Environmental factors are not tested in your paper. 

Line 68: “behavior” should be plural.

Response: Unnecessary information from the paragraph is removed and streamlined.

Line 103: I recommend adding “mitochondrial” in front of this first 16S to distinguish it from bacterial 16S once. After specifying this once, "16S rRNA" as short-hand is sufficient.

Response: Changed accordingly.

Line 110: Here you specify the randomized sampling of owned and stray dogs, yet in the methods, nothing is mentioned about the stray dogs. See Methods for further comments.

Response: We admit this is a typo and the word is removed. The stray dogs were not included to obtain the original picture of ticks infesting dogs otherwise if they were included then there is a chance of transfer or accidental attachment of ticks from other animals to dogs not known to infest them.

MATERIALS AND METHODS

Figure 1: What are “mixed” hair breeds? This is not defined in the text.

Response: Areas with mixed symbol means that for tick sampling both long and short-hair breeds we available.

Study design and tick sampling strategy

Lines 141-43: “Dogs of willing pet owners were included in this study”. In Line 110, you mentioned that both stray dogs and pet dogs were used. Here is where more information is needed. How many dogs were strays? how many were pets? How did you obtain information on the strays that could be used for comparison to the pets? Were these mark/recapture strays? Or are you referring to dogs that are considered pets, but have a roaming range that is not in a confined area (e.g. yard or home)? This needs clarification.

Response: The word “stray dogs” was a typo and was removed from the mentioned text in the introduction section. The dogs mentioned and considered here we all pets with a limited range of roaming (yard or home).

Lines 147-148: You mention two types of breeds: long hair and short hair, yet in the results, you mention medium hair (line 249) and also in Table 5. Please clarify or correct.

Line 152: what is a mouth gag? is that different from a muzzle?

Line 156: “shifted” change to “shipped”

Response: Mouth gag or muzzle are synonyms and included both in the text as gag/muzzle. The word shifted is changed to shipped.

Tick genomic DNA extraction and gene amplification: What was the purpose of using only 120 ticks (out of 1150) for molecular assay? If you wanted to confirm identification by assay for each location, wouldn’t you want to test a higher proportion by location, not 10% across the whole collection? Why did you assay 20 samples per taxon?  

Response: Actually, those tick samples were selected for further downstream analysis (genomic DNA extraction) that were morphologically intact and morpho-taxonomically identified up to species level (damaged or fully engorged samples were not considered). There was a fluctuation in the number of collected ticks from each sampling location (40 spots) ranging from 20-35 ticks/location). So, we selected 20 individual samples from each taxon to assess the accurate and unbiased profile of ticks and their-transmitted pathogens (Data not published yet) in the dog population of the study area. The tick samples were not pooled keeping in view the accurate detection of tick-borne pathogens in the dog population.

Line 168; Table 1: References for the primers are listed as the authors. Unless the sequences presented are incorrect, these sequences are the same as those from the source, Folmer et al 1994 and Black& Piesman 1994, respectively. If you had modified the primers in some way, perhaps citing yourself is acceptable, but as presented it is not.

Response: The points are noted the author self-citation substituted with original literature.

Amplicon purification and DNA sequencing

 Line 176: “MEGA VII” should be “MEGA7” (https://pubmed.ncbi.nlm.nih.gov/27004904/). The version after that was MEGA X, with a space and the Roman numeral, but the versions before and after were “MEGA7” and “MEGA11”, respectively (https://www.megasoftware.net/citations).

Response: The word changed according to the reviewer suggestion and thanks for additional information.

It looks like you sequenced 11 tick samples across taxa for the cytochrome oxidase subunit I and 12 for mitochondrial 16S rRNA. You have not specified from where these samples were collected, which is unfortunate since you had some information on district-specific tick data. 

Response: Yes, the observations are correct but actually we have sequenced all amplified products which has uniformly covered all tick sampling locations and the number of partially nucleotide sequences of cox1 and 16SrRNA genes reduced to 11 and 12 because these are consensus sequences to avoid redundancy in the phylogenetic trees and NCBI GenBank.

Phylogenetic analysis

Line 184 (see comment above)

Lines 182-189: In the Results section, you present 4 phylogenetic trees. The first two are Neighbor-joining trees that include all the sequences for the COI and 16S rRNA amplicons, respectively, but these have different outgroups (which you do not specify here). The outgroups in the results are not the same as those listed in one of the Figures (Figure 6 states that the outgroup for 16S is Psoroptes ovis, but the text in the Results section states that it is Sarcoptes scabiei). Please check for correctness.

Response: The relevant sentences about rooting of the phylogenetic trees were incorporated in the concerned section and the typo of outgroup was corrected accordingly.

The second pair of phylogenetic trees is analyzed with a Maximum Likelihood analysis, but the approach is slightly different (you did not include outgroups and it seems to be focused exclusively on the genus Rhipicephalus). Please explain why you did not use an outgroup here, but. you used one in the aforementioned tree. (Note that without an outgroup, you can discuss the clustering of sequences (relatedness within a group), but not the direction of evolution).

Response: The second pair of trees were constructed using ML algorithm and without use of outgroups as roots because here we only considered the brown dog tick Rh. sanguineus s.l.  to explore its clustering pattern and evolutionary relatedness with same isolates representing different linages (tropical and temperate).

Statistical analysis

Logistic regression is used, but not specified. What analysis/analyses? In the results, p-values are presented without any associated test statistics. Yes, the presence of confidence intervals is helpful, but if we don’t know from what analyses they were derived, it doesn’t mean anything. Are these also from the regression or some other analysis (t-test? chi-square? other? if it is a regression, do you have a correlation coefficient)? I suspect some explanatory paragraph was missed during the copy-paste/editing process, but it does need to be addressed. 

You used odds ratios in some of your results yet you do not discuss this here or in your results.

 Response:  Some of the points are addressed and remaining will be finalized very soon.

RESULTS

Host-based demographic Characteristics <--why is "Characteristics" capitalized"?

Response: This was typo and the title is streamlined to “Host demographic profile”.

Table 2: “List of reference primer sets used for the amplification of target genes of ticks” The title of this table is the same as Table 1. It is also cut off with headers on page 5 and the rest of the table on page 6. It is difficult to evaluate this table. I don’t know whether the numbers and percentages refer to the proportion of puppies of total dogs by district or the number/percentage of each subcategory with at least one tick by each district. As much of the data seems to be either unexplained or redundant in other figures/tables, perhaps add the data from Figure 2 into this table and remove Figure 2.

Response: Corrected the table heading and necessary changes inserted into the text and table. Fig. 2 is replaced with table.

Overall district

Prevalence = # dogs with at least one tick, abundance=# ticks per dog? I assume the prevalence, in this case, is the former, but it is not clear since you haven’t defined this anywhere. I assume this means that 183 of 300 dogs had at least one tick, but no information about co-infestation by species, number of ticks per dog(abundance), or number of species per dog characteristic is presented.  

Figure 2: Could be included in Table 2 (after it is fixed) or if presented as is, please add the N to the figure.

Response: Here we described the prevalence of ticks in dog population not tick abundance. The paragraph is rectified accordingly.

Overall and spatio-temporal distribution of tick species 

Figure 3: Please add N to the figure. Comment: Good of you for trying to use a more colorblind-friendly palette, but you might want to check on "Colorbrewer" (https://colorbrewer2.org) to make sure the colors are not too similar.

Temporal distribution: These were the data from one year, so it does not represent variation from year to year, but it is suggestive of seasonal trends.

Line 242: Another orphan subheading. “Host demographic-based prevalence of tick infestation in dog populations” should be moved to the next page.

Response: The suggested changes applied to the relevant section of the manuscript.

Lines 243-52: p-values are presented, but for what statistical analysis? CI is not an analysis.

Management factors-based prevalence of tick infestation in dogs 

Lines 255-264: Same comment for lines 243-52. The p-values are presented for what statistical analysis? There is a p-value missing for stray versus pet dogs. 

Line 255: “Environmental factors” which are what? It seems that you use “environmental” to be synonymous with “Management”, as listed in your subheading. Perhaps a copy-paste error?

Response: The “Environment” is replaced with “Management”.

Risk factors for the tick infestation 

Line 272-275: Font size is different

Response: Changes incorporated

Table 5. Risk factors of tick infestations of dog hosts.: An odd ratio is binary (yes or no, sick or not, dead or alive, infested or not, etc.). If you do have an odds ratio for multiple categories, you need an OR for each pairwise comparison. You need to clarify what you are presenting.

Line 277: Subheading “Molecular characteristics” This subheading should be moved to the next page. It looks like it was part of the “risk factors” table.

Line 278: “COICOI” should be “COI”

Response: subheading transferred to relevant place and gene name is rectified.

Phylogenetic analyses

Because there are two types of analyses presented here, you might want to have two subheadings: Phylogenetic analysis: Neighbor-joining and Phylogenetic analysis: Maximum Likelihood. It makes it easier to keep track of which analysis is being presented. 

Response: headings were incorporated accordingly as suggested.

Figures 5-8: General comments:

It might be helpful to include the district(s) from which the Pakistani R.s.s.l. sequences were derived. 

  The resolution is not great and some bootstrap values were difficult to read. Please make the bootstrap font size a bit larger (you can do this in MEGA before exporting the figure).

What was the actual fragment size after trimming? This should be in the figure legend and is provided by MEGA7 in the caption.

Figure 8: Egyptian specimens cluster with Clade III (the “south-eastern Europe lineage”) plus with another species. This broad clustering of Rssl plus three other species also occurred in Clade IV. If these were rooted in a sister taxon (e.g. Dermacentor) would these relationships be better resolved?

Line 320: “The 16S rRNA phylogenetic tree…” add “Neighbor-joining” before “phylogenetic” to distinguish this from the maximum likelihood analysis. 

Line 321 and Figure 6: Outgroups are not the same in the Figure legend as in the text or figure itself. Please correct.

Line 358: Suggested subheading: “Phylogenetic analysis - Maximum Likelihood”

Line 358: Not sure if “pertinence” is the right word. Perhaps “Rssl sequences from both COI and 16S rRNA amplicons from this study (Pakistan) cluster with the tropical lineage of Rssl (previously described by…)”.

As an aside, COI is the gold standard for barcoding but can be highly variable even within species (particularly in ticks). Thus, the use of mitochondrial 16S in conjunction with COI provides more robust support for your conclusions. Another point: phylogenies of mitochondrial sequences can also be misleading or biased, so ideally inclusion of a nuclear gene or one not under the same selection pressure is also preferred.

DISCUSSION

Lines 385-387: Period after “places of Pakistan [20, 22,28,29, 30,31]”. Then “One study compiled (catalogued?) tick species infesting dogs …, However, this study was …” 

Line 397: “We observed higher infestation levels (61%) on dogs from Pakistan than previously observed …”

**Note that your observations were for one year and do not reflect variability over time**

Line 405: “Risk factors associated with tick infestation are highly uneven between studies.” Perhaps “variable” in place of “uneven”? Is the unevenness referring to your studies or studies of ticks on dogs in general? 

Line 406-407: “female dogs were found with high(er?) tick infestation than male dogs, although no statistical difference was found”, but in univariate regression it was significant. This suggests that there are other factors contributing, but you do not address this. 

Line 407-416: You present two speculations here. The first is about females being more likely to get infested because they are taking care of puppies, but were all the females active mothers? This is not clear. The second was that puppies are in closer proximity to the ground and therefore have a higher likelihood of infestation than adults. This doesn’t make sense. They all have to walk on the ground and likely cover the same habitat. What you could have done (if there were enough data) was compare females that were with puppies versus females that were not with puppies and see if there was any correlation. 

Line 418-419: Reword this sentence. 

Line 424-5: Is it specific to the type of acaricide, or any use versus none? Was it known what type was used? (oral, topical, bath, spray, powder, etc). The frequency of treatment somewhere was mentioned as irregular or regular. How were those analyzed? 

Line 434: “gradual decline in July/August” —in some temperate species this corresponds to the decline of adults and the increase in nymphs, but not observed here. You suggest that the monsoon season may have contributed here. Line 439: High humidity is not the problem for ticks—they thrive in high humidity. “Increase in precipitation” makes sense based on the context of your sentence.

Line 455-46, “…continuous misidentification of most of the members of the complex is still on the rise” reword to “…complex still occurs”

Line 459: “woks” should be “works” 

Paragraph Lines 470-481 need some restructuring for flow:

Start with “Here we describe… a new canine association for Hy. excavatum. and H. dromedarii.” Hy excavatum ….” (Since you go into this at great lengths for Hy. dromedarii, you should probably also include geographic distribution for Hy. excavatum)

Next, describe Hy. dromedarii (hosts and geographic distribution: “Several studies reported hyenas, dogs, ostriches….” 

End with “We also present the first detailed molecular study on both of these species of ticks…”

Line 489: Note: Some of these Egyptian sequences clustered with the south-eastern Europe lineage. Can you discuss this?

Line 497-500: Be careful not to speculate too much about divergence—you can’t say much about evolutionary distance unless you root the tree.

Conclusion:

Line 502-507: Stop after “morphological identification”. Replace “however, it becomes inaccurate and sometimes misidentified without…” with “Reliance on morphological characteristics alone can result in misidentification by those with limited…”

Supplemental information:

Supplementary Tables 1 and 2: According to your phylogenetic tree and on Genbank the country is listed as “Pakistan” for these accession numbers, but in both tables, Rh.s.l. were listed as from China. You need to fix this.

 Response: The tables were corrected accordingly.

Reviewer 2 Report

Zeb et al. has submitted the paper entitled Ticks infesting dogs in Khyber Pakhtunkhwa, Pakistan: First epidemiological and molecular report for publication in the Pathogens.

I very humbly and kindly request to prominent scientists like Alejandro Cabezas-Cruz5, José de la Fuente6,7 and Olivier Andre Sparagano that tick fauna in Pakistan has been mostly misdiagnosed that have resulted into severe scientific issues specifically to young first-hand researchers in a country. I therefore, very kindly ask to be extremely careful while publishing papers that report ticks specifically from “Pakistan”. For instance:

Zeb et al is citing Rooman et al., PloS one. 2021;16 8:e0255138. Please open this paper and check they have reported Haemaphysalis aciculifer in Pakistan? Now I question the author how you cite such paper? Haemaphysalis aciculifer is a specific African tick! This is my major concern, while citing those papers which include very false data!

Another citation problem and false report from Pakistan: Zeb is citing Sultan et al. 10.1007/s00436-022-07596-3   - which report Hy marginatum from Pakistan? Are you sure Hy marginatum exist in Pakistan? I am sure no. This tick even does not exist along the borders of Pakistan with Afghanistan in study regions of the paper. I assume, please avoid such false data. I know this paper is from this group too. My suggestions are to avoid propagations of such false information through further citations.

Yet another example: Zeb is citing a paper from Pakistan “An assessment of the molecular diversity of ticks and tick-borne microorganisms of small ruminants in Pakistan” which molecularly report Haemaphysalis (Hs.) punctate from Pakistan, the sequences for this paper were also uploaded to genbank! e.g. accession number: MT800318.  The sequences uploaded from this paper is a big trouble for many researchers, you may search and find the reality. If you make the BLAST for these sequence uploaded for Haemaphysalis (Hs.) punctate from Pakistan, it will never ever hit the original Haemaphysalis (Hs.) punctate but other ticks. If papers have such blunder, I prefer to do not cite as to avoid creating further problems for young researchers who mostly lack the true in-depth knowledge. This also reflect the knowledge of the authors who cite such flaws full information.

Before doing an in-depth review I have severe issues with the falsification of originalities of scientific standards that are prominent in this paper.

1.      In the primers table 1, are these primers were designed by Zeb et al 2019? Why to do so if these were designed by Mangold et al. 1998 or Folmer et al. for the first time?

2.      line 23, is it truly first work? see my comment below. Karim et al. has done a better job in 2017.

3.      Line26, please check recent papers, this is not the first report. https://doi.org/10.3390/microorganisms10081663

4.      Line 449, is it true that you are R. sanguineus s.l for the first time from Pakistan? please check, https://doi.org/10.1371/journal.pntd.0005681, also, https://doi.org/10.1186/s13071-021-04836-w,  also https://doi.org/10.3389/fphys.2019.00793, also https://doi.org/10.3390/pathogens11020162 and many others. I think it is totally injustice not citing Karim et al., 2017, which put an initial step toward the ticks work in the last two decades – After Hoogstral and Dmitry.

5.      May I kindly ask the authors to please provide ticks pictures, this may clear the whole scenario? I will be glad to review this paper further if the authors provide Hy exacavtum pictures!

6.      there are several linguistic issues throughout the paper… Environmental factors-based prevalence showed significantly (P= 0.001) higher tick infestation (43.3%, CI= 37.7-48.9) was observed in dogs with no acaricidal application as compared to the irregular (12.7%, CI= 8.9-16.5) and regular use of acaricides (5.0%, CI= 2.5-7.5) (Table 4).

I am not recommending rejection of this paper, but I am waiting for the rectification of the paper as per above comments, or candid arguments from the authors. More specifically, I am waiting for Hy excavatum pictures. As I said above, Alejandro Cabezas-Cruz5, José de la Fuente6,7 and Olivier Andre Sparagano are very prominent scientists in the field who I truly respect. 

Author Response

Reviewer 2nd

Comments and Suggestions for Authors

Zeb et al. have submitted the paper entitled Ticks infesting dogs in Khyber Pakhtunkhwa, Pakistan: First epidemiological and molecular report for publication in the Pathogens.

I very humbly and kindly request to prominent scientists like Alejandro Cabezas-Cruz5, José de la Fuente6,7 , and Olivier Andre Sparagano that tick fauna in Pakistan has been mostly misdiagnosed that have resulted in severe scientific issues specific to young first-hand researchers in a country. I, therefore, very kindly ask you to be extremely careful while publishing papers that report ticks specifically from “Pakistan”. For instance:

Response: The misdiagnosed data is gradually rectified by the application of molecular approaches to avoid falsification in the scientific publications regarding ticks and TBDs across Pakistan as evident from the publications by Ali et al., Rehman et al., and Zeb et al.,

Zeb et al is citing Rooman et al., PloS one. 2021;16 8:e0255138. Please open this paper and check they have reported Haemaphysalis aciculifer in Pakistan. Now I question the author how you cite such paper. Haemaphysalis aciculifer is a specific African tick! This is my major concern while citing those papers which include very false data!

Response: your observations are noteworthy but it is for your kind consideration that the above-mentioned paper is cited on account of reporting R. sanguineus tick morpho-taxonomically. We have no concern with other tick species or falsification of the mentioned publication because it has been published by a reputable journal.

Another citation problem and false report from Pakistan: Zeb is citing Sultan et al. 10.1007/s00436-022-07596-3   - which report Hy marginatum from Pakistan? Are you sure Hy marginatum exist in Pakistan? I am sure no. This tick even does not exist along the borders of Pakistan with Afghanistan in study regions of the paper. I assume, please avoid such false data. I know this paper is from this group too. My suggestions are to avoid propagations of such false information through further citations.

Response: We are repeating our statement again that this paper has also documented the dog tick from its study area of interest. To date no renowned tick morpho-taxonomist is available in the country and most of the tick-associated research is done through reliable molecular techniques. Our current report is not dealing with Hyalomma marginatum however in response to your observations on the distribution of Hya. marginatum; this tick has already been reported from Baluchistan:

Kasi, K.K., von Arnim, F., Schulz, A., Rehman, A., Chudhary, A., Oneeb, M., Sas, M.A., Jamil, T., Maksimov, P., Sauter‐Louis, C. and Conraths, F.J., 2020. Crimean‐Congo hemorrhagic fever virus in ticks collected from livestock in Balochistan, Pakistan. Transboundary and Emerging Diseases67(4), pp.1543-1552. https://onlinelibrary.wiley.com/doi/full/10.1111/tbed.13488

Yet another example: Zeb is citing a paper from Pakistan “An assessment of the molecular diversity of ticks and tick-borne microorganisms of small ruminants in Pakistan” which molecularly report Haemaphysalis (Hs.) punctate from Pakistan, the sequences for this paper were also uploaded to genbank! e.g. accession number: MT800318.  The sequences uploaded from this paper is a big trouble for many researchers, you may search and find the reality. If you make the BLAST for these sequence uploaded for Haemaphysalis (Hs.) punctate from Pakistan, it will never ever hit the original Haemaphysalis (Hs.) punctate but other ticks. If papers have such blunder, I prefer to do not cite as to avoid creating further problems for young researchers who mostly lack the true in-depth knowledge. This also reflect the knowledge of the authors who cite such flaws full information.

Response: The research work is accurate without any troublesome regarding sequence hit on GenBank as this has been published by renowned and experts in this field i.e. Alejandro Cabezas-Cruz, Sara Moutailler, Robin B. Gasser and  Abdul Jabbar. The sequence: MT800318 hit its own and other similar isolates present in GenBank. For instance;

Before doing an in-depth review I have severe issues with the falsification of originalities of scientific standards that are prominent in this paper.

  1. In the primers table 1, are these primers were designed by Zeb et al 2019? Why to do so if these were designed by Mangold et al. 1998 or Folmer et al. for the first time?

Response: The point is noteworthy and raised also by R1. This was adopted from Lv et al., 2014 paper and the reference is replaced by the originalone.

  1. line 23, is it truly first work? see my comment below. Karim et al. has done a better job in 2017.
  2. Line26, please check recent papers, this is not the first report. https://doi.org/10.3390/microorganisms10081663

Response: Yes, this is truly first work on ticks infesting dogs across the study area. It is right that Kriam et al., 2017 had conducted a comprehensive study on Tick and Tick-borne livestock Pathogens in Pakistan but  1) His sampling host represents multiple species

       2) Identified ticks based on a morpho-taxonomic approach

       3) Specifically focused on TBDs of livestock and no in-depth detail regarding sampling host, its demography, etc. A number of recently published articles have reported new tick species from Pakistan, Which may be either missed or miss-diagnosed for instance: see the paper of tick and TBDs expert Dr. Abid Ali  who did a better job by reporting diverse tick fauna from Khyber Pakhtunkhwa Pakistan https://www.frontiersin.org/articles/10.3389/fphys.2019.00793/full

  1. Line 449, is it true that you are R. sanguineuss.l for the first time from Pakistan? please check, https://doi.org/10.1371/journal.pntd.0005681, also, https://doi.org/10.1186/s13071-021-04836-w,  also https://doi.org/10.3389/fphys.2019.00793, also https://doi.org/10.3390/pathogens11020162 and many others. I think it is totally injustice not citing Karim et al., 2017, which put an initial step toward the ticks work in the last two decades – After Hoogstral and Dmitry.

Response: Yes this is the first molecular report of R. sanguineus s./. from dogs in Pakistan. Off course, Karim et al., 2017 work is appreciable but we did not cite because of his focus on TBPs not tick specifically.

  1. May I kindly ask the authors to please provide ticks pictures, this may clear the whole scenario? I will be glad to review this paper further if the authors provide Hy exacavtum pictures!

Response: As we have done morpho-taxonomic based identification for the separation of ticks up to species level and further confirmed by molecular based identification which is more reliable than the previous one and we have submitted the sequences to the NCBI GenBank (16S rRNA ON921119, cox1 ON911979). We have not taken the tick photographs.

  1. there are several linguistic issues throughout the paper… Environmental factors-based prevalence showed significantly (P= 0.001) higher tick infestation (43.3%, CI= 37.7-48.9) was observed in dogs with no acaricidal application as compared to the irregular (12.7%, CI= 8.9-16.5) and regular use of acaricides (5.0%, CI= 2.5-7.5) (Table 4).

Response: The linguistic issues in the paper has reduced to minimum level and published version will be grammatical error free.

I am not recommending rejection of this paper, but I am waiting for the rectification of the paper as per above comments, or candid arguments from the authors. More specifically, I am waiting for Hy excavatum pictures. As I said above, Alejandro Cabezas-Cruz5, José de la Fuente6,7 and Olivier Andre Sparagano are very prominent scientists in the field who I truly respect. 

Response: The comments are thoroughly addressed and if need further clarification we shall be available.

Thank you.

Round 2

Reviewer 1 Report

To the Authors:

“Ticks infesting dogs in Khyber Pakhtunkhwa, Pakistan: First epidemiological and molecular report” is presented for review by Jehan Zeb et al. The authors present findings that include the presence of known and previously unknown species of ticks on dogs. They also present phylogenetic analyses of genetic data on a subset of each sample. The manuscript adds information known about tick abundance and species distribution of ticks found on domestic dogs in Pakistan. 

Most of the comments have been addressed except for the lack of test value (e.g. chi? t-test, F or Z statistic, etc.) associated with the p-values. There are some minor editing comments listed below that can be fixed by you or the copy editor. 

Specific areas of improvement are listed below:

ABSTRACT

L26: “Cytochrome Oxidase 1” should be Cytochrome Oxidase subunit I”. This is correct in your Introduction.

L27: “rRNA gene”. You sequenced the gene that encodes the rRNA, not the rRNA itself. It can be abbreviated to 16S rRNA subsequently.

L40-43: Check the species epithets. They were autocorrected to capital letters.

L47: remove the ellipse and replace it with a single period.

L49: “Stray dogs” are mentioned here, but as pet dogs in the Introduction (L104) and Methods(L135-6). 

L51: “on dongs” —>“on dogs”

INTRODUCTION

L96: “subunit I”. From here you can use cox1 since you’ve defined it.

METHODS

L140: One sentence should include the question of whether dogs are free-roaming or not.  You mention it in the results and as one of the categories in Table 4 but should be included in the methods first.

Phylogenetic analysis: references needed for models. MEGA7 can provide them when you generate your tree

L192 “Statistical analysis of emperical data” <—Should be “empirical”

RESULTS

L272-274: “Potential risk factors for the tick infestation” should be on one line

L286: “cox1and” needs a space

p-values are presented without any associated test statistics. CI is not an analysis.

Table 5. Risk factors of tick infestations of dog hosts.: Odds ratio is binary. I think in the text you group puppy and juvenile versus adult, but the OR for acaricide application, does it refer to no versus any or regular versus irregular/none?

Figures 5-8: General comments:

The figure resolution is low and the bootstrap values are difficult to read. Please make the bootstrap font size larger. Fragment size after trimming should be included in the legend. (MEGA7, MEGA X, or MEGA11 can generate a better resolution image, references for models/analyses, and fragment size).

L 299: “cox1sequences” needs a space.

L308: “cox1and” needs a space

L317: Should read “Ticks of the genus Rhipicephalus associated with ruminants were characterized…” Otherwise, it reads that the ruminants are of the genus “Rhipicephalus”

DISCUSSION

L426-429: “ female dogs were found with high tick infestation than male dogs, although no statistical difference was found. This could be possible that female dogs have a sedentary lifestyle while nursing their puppies, which allows ticks to infest them [45].”

L432: “t adult dogs” <—remove “t”

L434: change semi-colon to colon

L434-436:  Were all the females actively nursing mothers? Were puppies more likely to be infested than mothers? The suggestion that puppies are closer to the ground doesn’t make sense if they travel/inhabit the same habitat. 

SUPPLEMENTARY TABLES: This has not yet been addressed.

Supplementary Tables 1 and 2: According to your phylogenetic tree and on Genbank the country is listed as “Pakistan” for these accession numbers, but in both tables, Rh.s.l. were listed as from China.  Please explain or fix it.

Author Response

Dear Reviewer 1,

Thank you very much for your constructive comments on our manuscript which have improved our article contents.

Comments and Suggestions for Authors

Reviewer 1

To the Authors:

Most of the comments have been addressed except for the lack of test value (e.g.chi? ttest, F or Z statistic, etc.) associated with the p-values. There are some minor editing comments listed below that can be fixed by you or the copy editor.

Response: The relevant statistics were added in the concerned tables after revision.

Specific areas of improvement are listed below:

ABSTRACT

L26: “Cytochrome Oxidase 1” should be Cytochrome Oxidase subunit I”. This is correct in
your Introduction.

Response: Changed accordingly.

L27: “rRNA gene”. You sequenced the gene that encodes the rRNA, not the rRNA itself. It
can be abbreviated to 16S rRNA subsequently.

Response: the typo is corrected.

L40-43: Check the species epithets. They were autocorrected to capital letters.

Response: The species epithets were revised and rectified.

L47: remove the ellipse and replace it with a single period.

Response: Changed accordingly.

L49: “Stray dogs” are mentioned here, but as pet dogs in the Introduction (L104) and
Methods(L135-6).

Response: The mentioned word is replaced with correct words.

L51: “on dongs” —>“on dogs”

Response: Changed

INTRODUCTION
L96: “subunit I”. From here you can use cox1 since you’ve defined it.
Response: replaced with cox1

METHODS
L140: One sentence should include the question of whether dogs are free-roaming or not.
You mention it in the results and as one of the categories in Table 4 but should be
included in the methods first.

Response: Information added in the Methodology and in the results.

Phylogenetic analysis: references needed for models. MEGA7 can provide them when
you generate your tree

Response: Relevant references were added.

L192 “Statistical analysis of emperical data” <—Should be “empirical”

Response: Changed

RESULTS
L272-274: “Potential risk factors for the tick infestation” should be on one line.

Response: Shifted to proper place.

L286: “cox1and” needs a space

Response: Typo removed.

p-values are presented without any associated test statistics. CI is not an analysis. Table 5. Risk factors of tick infestations of dog hosts.: Odds ratio is binary. I think in the text you group puppy and juvenile versus adult, but the OR for acaricide application, does it refer to no versus any or regular versus irregular/none?

Response: All the relevant statistical values were added and clarified for the readers.

Figures 5-8: General comments: The figure resolution is low and the bootstrap values are difficult to read. Please make the bootstrap font size larger. Fragment size after trimming should be included in the legend. (MEGA7, MEGA X, or MEGA11 can generate a better resolution image, references for models/analyses, and fragment size).

Response: All the figures were replaced with good resolution figures. The study area map is replaced with new map figure which is clearer and more informative about the host population and collected tick species.

L 299: “cox1sequences” needs a space.

Response: Changed.

L308: “cox1and” needs a space

Response: Changed.

L317: Should read “Ticks of the genus Rhipicephalus associated with ruminants were characterized…” Otherwise, it reads that the ruminants are of the genus “Rhipicephalus”.

Response: Revised the sentence to clarify the piece of information for readers.

DISCUSSION
L426-429: “female dogs were found with high tick infestation than male dogs, although no statistical difference was found. This could be possible that female dogs have a sedentary lifestyle while nursing their puppies, which allows ticks to infest them [45].”

Response: Revised the sentence and relevant references were added.

L432: “t adult dogs” <—remove “t”

Response: Changed.

L434: change semi-colon to colon

Response: Changed.

L434-436: Were all the females actively nursing mothers? Were puppies more likely to be infested than mothers? The suggestion that puppies are closer to the ground doesn’t make sense if they travel/inhabit the same habitat.

Response: Revised the sentence and relevant references were added.

SUPPLEMENTARY TABLES: This has not yet been addressed.
Supplementary Tables 1 and 2: According to your phylogenetic tree and on GenBank the
country is listed as “Pakistan” for these accession numbers, but in both tables, Rh.s.l.
were listed as from China. Please explain or fix it.

Response: Revised the supplementary tables as suggested to remove the typos.

Reviewer 2 Report

Hello, I think authors were unable to rectify the paper as per my suggestions. 

Author Response

Comments and Suggestions for Authors

Reviewer 2

To the Authors:

Dear Reviewer 2,

Your comments were constructive and has improved the final version of the manuscript.

In response to your comments form;

The manuscript passed twice through major revision. The necessary changes were incorporated successfully as suggested. We have addressed all the points raised during review.